# Type I and II PRMTs inversely regulate post-transcriptional intron detention through Sm and CHTOP methylation

**Maxim I Maron[1], Alyssa D Casill[2], Varun Gupta[3], Jacob S Roth[1], Simone Sidoli[1], Charles C Query[3], Matthew J Gamble[2,3], David Shechter[1]\***

[1]Department of Biochemistry, Albert Einstein College of Medicine, Bronx, United States; [2]Department of Molecular Pharmacology, Albert Einstein College of Medicine, Bronx, United States; [3]Department of Cell Biology, Albert Einstein College of Medicine, Bronx, United States

**\*For correspondence:**
david.shechter@einsteinmed.edu

**Competing interest:** The authors declare that no competing interests exist.

**Abstract** Protein arginine methyltransferases (PRMTs) are required for the regulation of RNA processing factors. Type I PRMT enzymes catalyze mono- and asymmetric dimethylation; Type II enzymes catalyze mono- and symmetric dimethylation. To understand the specific mechanisms of PRMT activity in splicing regulation, we inhibited Type I and II PRMTs and probed their transcriptomic consequences. Using the newly developed Splicing Kinetics and Transcript Elongation Rates by Sequencing (SKaTER-seq) method, analysis of co-transcriptional splicing demonstrated that PRMT inhibition resulted in altered splicing rates. Surprisingly, co-transcriptional splicing kinetics did not correlate with final changes in splicing of polyadenylated RNA. This was particularly true for retained introns (RI). By using actinomycin D to inhibit ongoing transcription, we determined that PRMTs post-transcriptionally regulate RI. Subsequent proteomic analysis of both PRMT-inhibited chromatin and chromatin-associated polyadenylated RNA identified altered binding of many proteins, including the Type I substrate, CHTOP, and the Type II substrate, SmB. Targeted mutagenesis of all methylarginine sites in SmD3, SmB, and SmD1 recapitulated splicing changes seen with Type II PRMT inhibition, without disrupting snRNP assembly. Similarly, mutagenesis of all methylarginine sites in CHTOP recapitulated the splicing changes seen with Type I PRMT inhibition. Examination of subcellular fractions further revealed that RI were enriched in the nucleoplasm and chromatin. Taken together, these data demonstrate that, through Sm and CHTOP arginine methylation, PRMTs regulate the post-transcriptional processing of nuclear, detained introns.

## Introduction

The mammalian genome encodes nine protein arginine methyltransferases (PRMTs 1–9; PRMT4 is also known as CARM1). Arginine methylation is critical in regulating signal transduction, gene expression, and splicing (reviewed by *Guccione and Richard, 2019*; *Lorton and Shechter, 2019*). For instance, an important function of PRMT5 and its cofactors pICln and MEP50 (also known as WDR77) is assembly of small nuclear ribonucleoproteins (snRNPs)—core components of the spliceosome (*Meister et al., 2001*; *Neuenkirchen et al., 2015*). This includes both non-enzymatic chaperoning of Sm proteins via PRMT5/pICln following their translation and post-translational methylation of SmD1 (SNRPD1), SmD3 (SNRPD3), and SmB/B′ (SNRPB), by PRMT5-MEP50 (*Friesen et al., 2001*; *Meister et al., 2001*; *Paknia et al., 2016*). Following their methylation, these Sm proteins are delivered to the Survival of Motor Neuron (SMN) assembly factor, by which they are bound to small nuclear RNAs (snRNAs) in preparation for further processing and eventual nuclear import (*Boisvert et al., 2002*; *Meister and Fischer, 2002*; *Pellizzoni et al., 2002*). Disruption of PRMT5 leads to numerous splicing defects,

primarily exon skipping (SE) and intron retention (RI) (*Boisvert et al., 2002*; *Bezzi et al., 2013*; *Koh et al., 2015*; *Braun et al., 2017*; *Fedoriw et al., 2019*; *Fong et al., 2019*; *Radzisheuskaya et al., 2019*; *Tan et al., 2019*; *Li et al., 2021*; *Sachamitr et al., 2021*). Intron retention is a highly prevalent alternative splicing event in tumors and is ubiquitous across cancer types (*Dvinge and Bradley, 2015*). Therefore, determining how PRMT activity governs intron retention is of great interest.

RNA splicing occurs either during transcriptional elongation or after the transcript has been cleaved and released from polymerase (reviewed by *Neugebauer, 2019*). Although the importance of PRMT5 in preserving splicing integrity is clear, whether PRMT5 exerts its influence over co- or post-transcriptional splicing is still unknown. Previous work has indirectly implicated PRMT5 in the regulation of post-transcriptional splicing in *Arabidopsis* and also as a regulator of detained introns (DI)—a class of retained, unspliced introns in poly(A) RNA that remain nuclear and are removed prior to cytoplasmic export (*Braun et al., 2017*; *Jia et al., 2020*). DI are highly conserved and enriched in RNA processing factor transcripts; their removal has been proposed as part of a feedback mechanism to control protein expression during differentiation and cell stress (*Yap et al., 2012*; *Wong et al., 2013*; *Braunschweig et al., 2014*; *Boutz et al., 2015*; *Pimentel et al., 2016*). As PRMT5 inhibition leads to loss of Sm methylation and a simultaneous increase in DI, Sm proteins have been implicated in regulating DI (*Braun et al., 2017*). However, a direct connection of DI or RI levels to Sm arginine methylation has not been established.

The mechanistic role of Type I PRMTs (PRMT1–4, 6, and 8) in splicing is still poorly understood. Recent reports demonstrate that Type I PRMT inhibition leads to SE consequences (*Fedoriw et al., 2019*; *Fong et al., 2019*). Importantly, many of the Type I-methylated proteins have a potential role in RNA post-transcriptional regulation (*Fedoriw et al., 2019*; *Fong et al., 2019*). Consistent with a role in post-transcriptional processing, the primary Type I methyltransferase, PRMT1, has been implicated in the regulation of RNA export; this regulation occurs through methylation of the Transcription and Export (TREX) components—Aly and REF export factor (ALYREF) and chromatin target of PRMT1 (CHTOP) (*Hung et al., 2010*; *van Dijk et al., 2010*; *Chang et al., 2013*). Loss of CHTOP leads to an accumulation of nuclear polyadenylated transcripts, consistent with a block in RNA export (*Chang et al., 2013*). Furthermore, ALYREF and CHTOP are both deposited along nascent RNA and are also bound to RI (*Viphakone et al., 2019*). As the nuclear export of RNA is intimately linked to splicing— dissociation of splicing factors is a prerequisite for successful export—regulation of this process is a potential means to control RI (*Luo and Reed, 1999*). Despite this, the role of nuclear export factors and the impact of Type I PRMTs on regulating RI remains unexplored.

Here, we test both the co- and post-transcriptional consequences of inhibiting Type I and II PRMTs. We also probe which proteins mediate the transcriptional consequences of PRMT inhibition. As both Type I and II PRMTs have been extensively reported to be required for the pathogenicity of lung cancer (*Hwang et al., 2021*)—yet their role in the transcription and splicing of this disease remains largely unstudied—we used A549 human alveolar adenocarcinoma cells as our model. Using GSK591 (also known as EPZ015866 or GSK3203591), a potent and selective inhibitor of PRMT5 (*Duncan et al., 2016*), and MS023, a potent pan-Type I inhibitor (*Eram et al., 2016*), we demonstrate that PRMTs inversely regulate RI post-transcriptionally through either Sm or CHTOP arginine methylation.

## Results
### Type I and II PRMT inhibition promotes changes in alternative splicing

PRMTs consume S-adenosyl methionine (SAM) and produce S-adenosyl homocysteine (SAH) to catalyze the post-translational methylation of either one or both terminal nitrogen atoms of the guanidino group of arginine (*Figure 1a*; *Gary and Clarke, 1998*). All PRMTs can generate monomethyl arginine (Rme1). Type I PRMTs further catalyze the formation of asymmetric $N^G,N^G$-dimethylarginine (Rme2a); Type II PRMTs (PRMT5 and 9) form symmetric $N^G,N'^G$-dimethylarginine (Rme2s). PRMT5 is the primary Type II methyltransferase (*Yang et al., 2015*). As previous reports indicated that lengthy treatment with PRMT inhibitors promotes aberrant RNA splicing, we wanted to determine whether alternative splicing differences occurred as early as day 2 and, if so, how they changed over time (*Bezzi et al., 2013*; *Fong et al., 2019*; *Radzisheuskaya et al., 2019*; *Tan et al., 2019*; *Li et al., 2021*; *Sachamitr et al., 2021*). Therefore, we performed poly(A)-RNA sequencing on A549 cells treated with DMSO, GSK591, MS023, or both inhibitors in combination for 2, 4, and 7 days. As 7 days of co-treatment

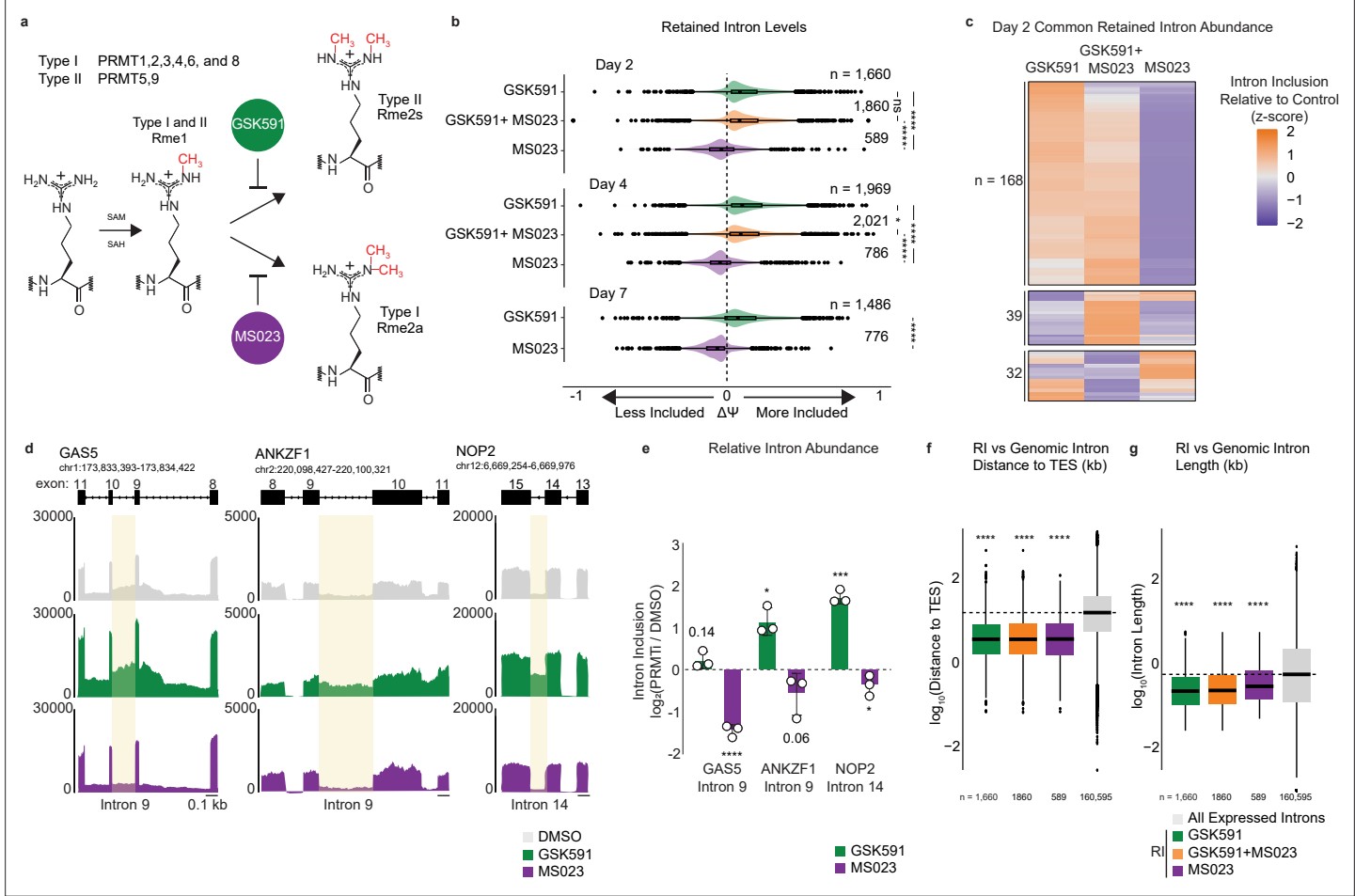

**Figure 1.** Type I PRMTs and PRMT5 inversely regulate intron retention. (**a**) Overview of protein arginine methyltransferases and their catalyzed reactions. (**b**) Comparison of $\Delta\Psi$ for retained introns (RI) following PRMT inhibition where $\Delta\Psi = \Psi$ (PRMT inhibitor)$-\Psi$ (DMSO). Significance determined using Kolmogorov-Smirnov test; *<0.05, ****<0.0001, ns=not significant. (**c**) Comparison of $\Delta\Psi$ z-score for common RI after 2-day treatment with PRMT inhibitors. (**d**) Genome browser track of poly(A)-RNA seq aligned reads for *GAS5, ANKZF1,* and *NOP2.* Yellow shading denotes RI. Scale (0.1 kb) indicated in lower right corner. (**e**) RT-qPCR of RI highlighted in panel (**d**). Data are represented as mean ± SD. (**f, g**) Comparison of RI and A549-expressed intron distance to the transcription-end site (TES) (**f**) or intron length (**g**) in $\log_{10}$(kb). Dashed line indicates genomic median; solid line within boxplot is condition-specific median. Significance determined using Wilcoxon rank-sum test; ****<0.0001.

The online version of this article includes the following source data and figure supplement(s) for figure 1:

**Figure supplement 1.** PRMTkd recapitulates RI inclusion seen with PRMT inhibition.

**Figure supplement 1—source data 1.** Western blot data for *Figure 1—figure supplement 1c*.

**Figure supplement 1—source data 2.** Western blot data for *Figure 1—figure supplement 1d*.

**Figure supplement 2.** PRMTs regulate a conserved class of retained introns (RI) that share unique features.

resulted in significant cell death, these cells were not sequenced (not shown). Using replicate Multivariate Analysis of Transcript Splicing (rMATS) (*Shen et al., 2014*) to identify alternative splicing events, we observed significant differences in RI with GSK591, MS023, and co-treatment with both inhibitors (FDR < 0.05) relative to DMSO (*Figure 1b*). Furthermore, whereas RI were increased with GSK591 and co-treatment—signified by a positive difference in percent spliced in (+ΔPSI or +$\Delta\Psi$)—RI were decreased with MS023 treatment (−$\Delta\Psi$) relative to DMSO.

We next intersected all common RI between treatment conditions in A549 at day 2 (*Figure 1c*). We found 239 common introns. Strikingly, whereas GSK591 and co-treatment with both inhibitors had increased inclusion, for the same introns, MS023 treatment resulted in decreased inclusion relative to DMSO (*Figure 1c*). When analyzing the distribution of $\Delta\Psi$ for this subset of common introns (as in *Figure 1b*), the median $\Delta\Psi$ did not significantly differ from the parent population (*not shown*). This

observation indicates that this subset of RI is suitably representative of the larger group of PRMT-regulated RI. To confirm and quantify the presence of PRMT-regulated RI by RT-qPCR, we selected three representative genes, *GAS5*, *ANKZF1*, and *NOP2*. These were identified by rMATS as containing inversely regulated RI after either Type I or II PRMT inhibition (*Figure 1d*). Notably, while the expression of *GAS5* and *NOP2* was increased with GSK591 treatment, it was unchanged with MS023 treatment. Using primers that spanned the intron-exon boundaries of *GAS5* intron 9, *ANKZF1* intron 9, and *NOP2* intron 14 and normalized by exonic primers, we confirmed the presence of the RI at day 2 of GSK591 or MS023 treatment (*Figure 1e*). Consistent with our rMATS data, *GAS5* intron 9, *ANKZF1* intron 9, and *NOP2* intron 14 had increased retention upon GSK591 treatment (p=0.141, 0.016, and 0.001, respectively). Furthermore, despite decreased dynamic range inherent to quantifying intronic signal loss, we observed less retention with MS023 treatment (p<0.0001, 0.064, and 0.040, respectively) (*Figure 1e*).

To confirm that these RI were specific to the PRMT inhibitors and not a consequence of off-target effects of GSK591 or MS023, we generated A549 cells expressing dCas9-KRAB-MeCP2 (*Yeo et al., 2018*). Independent transduction of two unique guide RNAs (gRNA) for 4 days targeting either the major Type I methyltransferases—PRMT1 and PRMT4—or the Type II methyltransferase, PRMT5, resulted in a robust reduction of their RNA expression (*Figure 1—figure supplement 1a*). Furthermore, we recapitulated increased RI in *GAS5* intron 9, *ANKZF1* intron 9, and *NOP2* intron 14 by RT-qPCR following knockdown of PRMT5 with both unique gRNAs (*Figure 1—figure supplement 1b*). When analyzing RI following transduction with the stronger PRMT1 gRNA, we also observed reduced RI in *GAS5* intron 9, *ANKZF1* intron 9, but not *NOP2* intron 14 (*Figure 1—figure supplement 1b*). Knockdown of PRMT4 only reduced RI in *GAS5* intron 9 with one gRNA, but not in *ANKZF1* intron 9 or *NOP2* intron 14. As the effect of the PRMT knockdown on RI levels was mild relative to inhibitor treatment, we next compared cellular methylarginine levels between the gRNA transduced cells to inhibitor treated cells (*Figure 1—figure supplement 1c*). Consistently, treatment with GSK591 or MS023 resulted in a stronger decrease in total cellular dimethylarginine levels, while PRMT knockdown only modestly decreased dimethylarginine, likely owing to incomplete depletion of the enzymes through this approach.

We recently published an extensive proteomic and transcriptomic characterization of the A549 arginine methylome in response to GSK591 and MS023 treatment (*Maron et al., 2021*). In that work, we identified 2444 unique methylations across 585 proteins, the majority of which were involved in nuclear and chromatin regulation. Therefore, to increase our signal of methylarginine changes following PRMT inhibition, we specifically analyzed the chromatin fraction (*Figure 1—figure supplement 1d*). We observed the following changes: (1) an increase in Rme1 and Rme2s with MS023 treatment; (2) an increase in Rme2a with GSK591 treatment; (3) decreased Rme2s with GSK591 treatment; and (4) decreased Rme2a with MS023 treatment. In addition to previously published reports using these same inhibitors, our data support that GSK591 and MS023 are specific and effective in depleting cellular methylarginine (*Chan-Penebre et al., 2015*; *Duncan et al., 2016*; *Eram et al., 2016*; *Fong et al., 2019*; *Plotnikov et al., 2020*; *Maron et al., 2021*; *Sachamitr et al., 2021*). As MS023 inhibits multiple Type I enzymes, the consequences of MS023 treatment on RI are likely more robust than the targeted CRISPR interference due to the ability of the remaining PRMTs to scavenge each other's substrates (*Dhar et al., 2013*; *Eram et al., 2016*; *Maron et al., 2021*).

## PRMT-regulated RI are conserved across cell types and share common characteristics

We next asked whether the RI in our data were common to other data sets in which PRMT activity was perturbed. To accomplish this, we used rMATS on publicly available data (*Braun et al., 2017*; *Fedoriw et al., 2019*; *Fong et al., 2019*; *Radzisheuskaya et al., 2019*). In these experiments—conducted in a variety of cell lines from diseases including acute myeloid leukemia (THP-1), chronic myeloid leukemia (K562), pancreatic adenocarcinoma (PANC03.27), and glioblastoma (U87)—arginine methylation was inhibited via PRMT knockdown using CRISPRi or with chemical inhibition using GSK591 or MS023. As demonstrated by the high odds ratio ($\log_2(OR)>6$) between all the data sets, we showed that there was a highly significant overlap in RI (Fisher's exact adjusted p<1e−05, *Figure 1—figure supplement 2a*). This high level of overlap between cell models and inhibition approaches further establishes the

importance of PRMTs in regulating RI and also supports that the observed changes in RI are on-target effects of the inhibitors.

RI have previously been reported to have common characteristics such as being shorter in length, closer to transcription end site (TES), and having reduced splice site strength (*Braunschweig et al., 2014*). To determine the common characteristics of the PRMT-regulated RI, we analyzed their length, distances to the TES, and sequences. We found that the RI were significantly closer to the TES and shorter when compared to the genomic distribution of A549 expressed introns (p<2.2e−16) (*Figure 1f and g*). Moreover, in analyzing the probability of nucleotide distribution at the 5′ and 3′ splice sites, we noted both a preference for guanine three nucleotides downstream of the 5′ splice site and increased frequency of cytosine in the polypyrimidine tract (*Figure 1—figure supplement 2b*). This is consistent with previous literature demonstrating that RI have common features contributing to their persistence in poly(A) RNA (*Bezzi et al., 2013*; *Braunschweig et al., 2014*; *Braun et al., 2017*; *Tan et al., 2019*). Despite GSK591 and MS023 treatments' inverse effect on RI, these characteristics did not differ between the observed retained or removed introns. Therefore, to uncover the mechanisms behind our observed splicing changes, we sought to characterize their co-transcriptional consequences.

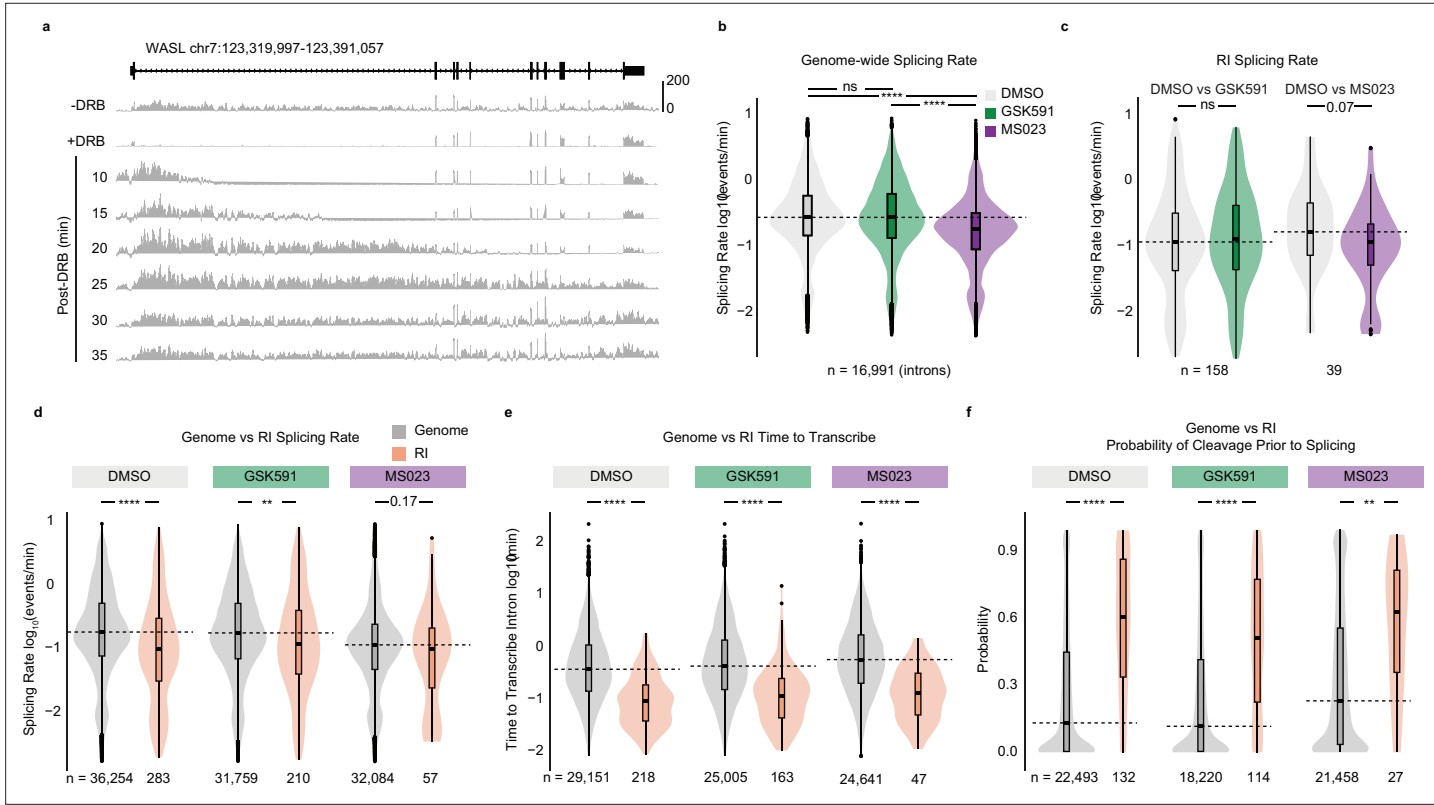

**Figure 2.** Retained introns (RI) share unique characteristics and are independent of PRMT-regulated co-transcriptional splicing. (**a**) Histogram of SKaTER-seq aligned reads across *WASL*. (**b**) Distribution of splicing rates for common genomic introns. Dashed line indicates DMSO median; solid line within boxplot is condition-specific median. Significance determined using Wilcoxon rank-sum test; ****<0.0001, ns=not significant. (**c**) Distribution of splicing rates for common RI as in panel (**b**). (**d–f**) Distribution of splicing rates (**d**), time to transcribe (**e**), and the probability of cleavage prior to splicing (**f**) for RI (orange) and A549 expressed introns (dark gray) within the same condition. Dashed line indicates genomic median; solid line within boxplot is condition-specific median. Significance determined using Wilcoxon rank-sum test; *<0.05, **<0.01, ***<0.001, ****<0.0001.

The online version of this article includes the following figure supplement(s) for figure 2:

**Figure supplement 1.** Transcript spawn rate correlates with expression and alternative splicing events are slower than constitutive ones.

**Figure supplement 2.** Splicing rate and probability of cleavage prior to splicing correlate with intron position.

## PRMT-dependent changes in co-transcriptional splicing do not reflect changes in poly(A) RNA

The inverse relationship on RI by Type I or II inhibition in addition to their TES-proximal location, shorter length, and non-canonical splice sites led us to specifically investigate the kinetics of co-transcriptional splicing. To accomplish this, we used Splicing Kinetics and Transcript Elongation Rates by Sequencing (SKaTER seq) (*Casill et al., 2021*). A brief overview of the method follows. To synchronize transcription, SKaTER seq uses a 3-hr 5,6-dichloro-1-β-D-ribofuranosylbenzimidazole (DRB) treatment, followed by a rapid washout to allow productive elongation to commence. Once RNA pol II begins elongating, nascent RNA is collected every 5 min until 35 min post-DRB washout. Nascent RNA is isolated via a 1 M urea wash of chromatin and an additional poly(A)-RNA depletion. Following sequencing, the rate of nascent RNA formation—including: (1) RNA pol II initiation and pause-release (spawn) rate, (2) elongation rate, (3) splicing rate, and (4) transcript cleavage rate—is calculated using a comprehensive model that determines the rates that best fit the sequencing coverage (*Casill et al., 2021*).

As splicing changes are present 2 days following GSK591 or MS023 treatment, we performed SKaTER seq at this treatment time point (*Figure 2a*). To assess the accuracy of the rates determined by the SKaTER model, we used the spawn, elongation, splicing, and cleavage rates to simulate a predicted poly(A)-RNA cassette exon $\Psi$ and compared these results with our poly(A)-RNA sequencing. We successfully predicted cassette exon $\Psi$ detected in poly(A) RNA in DMSO (Spearman's correlation coefficient $\rho$ =0.54), GSK591 ($\rho$ =0.61), and MS023 ($\rho$ =0.55) (p<2.2e−16) treatment conditions (*Figure 2—figure supplement 1a*). Next, we compared spawn rate to poly(A)-RNA sequencing transcripts per million (TPM). As expected, we observed a strongly significant (p<2.2e−16) correlation between RNA pol II spawn rate and TPM in DMSO-, GSK591-, and MS023-treated cells ($\rho$ =0.50, 0.51, and 0.51, respectively) (*Figure 2—figure supplement 1b*; *Casill et al., 2021*). Consistent with previously published reports, a comparison of splicing rates within each condition confirmed that constitutive introns splice faster than cassette exons and also faster than alternative 5′ and 3′ splice sites (*Figure 2—figure supplement 1c*; *Pandya-Jones et al., 2013*).

Poly(A)-RNA sequencing had revealed opposing splicing changes with PRMT inhibition: increased RI with GSK591 treatment and decreased RI with MS023 treatment (*Figure 1b*). To determine if this was a direct result of altered splicing rates, we used the SKaTER-seq data to ask how inhibiting PRMTs affected the genome-wide distribution of splicing rates. Surprisingly, compared to the DMSO control, GSK591 treatment did not significantly change the median of this distribution (Wilcoxon rank-sum test, p>0.05), while MS023 treatment resulted in significantly slower global splicing rates (p<2.2e−16) (*Figure 2b*). We next compared the splicing rates of introns that were retained with PRMT inhibition to the same introns in DMSO-treated cells. We found no significant difference in the median of GSK591-treated cells compared to DMSO (p=0.53), while MS023 treatment led to slower splicing rates (p=0.07) (*Figure 2c*). As we initially hypothesized that a slower splicing rate should increase RI, this result was surprisingly inconsistent with the PRMT-inhibited changes seen in poly(A) RNA (*Figure 1b*). These observations therefore suggested that changes in intron retention following PRMT inhibition were not entirely due to co-transcriptional splicing.

## RI share unique characteristics that limit their co-transcriptional removal

We next asked how RI splicing rates compared to non-RI. We observed that, when compared to all introns, introns that were retained in GSK591- or MS023-treated cells were slower to splice (p=0.01 and p=0.17, respectively) (*Figure 2d*). However, consistent with slower splicing as a general feature of RI, analysis of these same RI in DMSO-treated cells similarly revealed a significantly slower splicing rate (p=3.6e−6) (*Figure 2d*, left violin plots). As we previously found that RI were more likely to be closer to the TES (*Figure 1f*), we asked whether intron position correlated with splicing rate. Indeed, we determined that introns located closer to the TES had slower splicing rates (*Figure 2—figure supplement 2a*). Importantly, we had also previously found that RI were significantly shorter than non-RI (*Figure 1g*). The time required to transcribe an intron is a determining factor of splice site availability and co-transcriptional splicing outcomes (*Fong et al., 2014*). Thus, we next analyzed the time required to transcribe RI in comparison to non-RI. We found that RI transcription was completed significantly faster in DMSO-, GSK591-, and MS023-treated cells (p=1.03e−24, 1.94e−15,

and 3.00e−7, respectively) (*Figure 2e*). This observation, together with the proximity of RI to the TES and their slower splicing rate, suggests that RI in all conditions have a decreased window of opportunity for splicing to occur co-transcriptionally. Moreover, the lack of specific co-transcriptional splicing changes in RI following PRMT inhibition further supports that a post-transcriptional mechanism underlies the observed PRMT-dependent changes in poly(A) RNA.

The paradox that—despite their slower co-transcriptional splicing rate—RI are decreased with Type I PRMT inhibition led us to hypothesize that PRMTs exert their control over splicing post-transcriptionally. To test this hypothesis, we used the elongation, splicing, and cleavage rates determined by SKaTER seq to calculate the probability that a transcript will be cleaved from RNA pol II prior to a given intron being spliced co-transcriptionally. Consistent with most splicing being co-transcriptional, we found that the median probability of cleavage prior to splicing genome-wide was 9.7% in DMSO-, 8.7% in GSK591-, and 18.1% in MS023-treated cells (*Figure 2—figure supplement 2b*). The global distribution was significantly reduced with GSK591 treatment (p=0.002) and increased with MS023 treatment (p<2.2e−16) when compared to DMSO treatment. We next analyzed the probability of cleavage prior to splicing for RI compared to non-RI. Consistent with the hypothesis that PRMTs regulate RI post-transcriptionally, the median probability of transcript cleavage prior to co-transcriptional splicing was significantly higher for RI in DMSO- (p=2.65e-14), GSK591- (p=2.70e-9), and MS023-treated (p=0.004) cells (*Figure 2f*). Furthermore, intron position was strongly predictive of whether transcript cleavage was likely to occur prior to splicing: TES proximal introns had a higher probability of cleavage prior to their being spliced (*Figure 2—figure supplement 2c*). Altogether, the proximity of RI to the TES, shorter length, and slower splicing rate likely drives their decreased probability of being spliced co-transcriptionally.

## PRMTs regulate RI post-transcriptionally

The lack of evidence supporting a co-transcriptional mechanism for PRMTs in regulating RI led us to hypothesize that RI are regulated post-transcriptionally. Specifically, we hypothesized that MS023 promotes more efficient post-transcriptional intron decay, whereas GSK591 promotes less efficient post-transcriptional intron decay. To test this model, we pre-treated A549 cells for 2 days with GSK591 or MS023 and then blocked transcription using actinomycin D (ActD) for 60 min. We then performed poly(A)-RNA sequencing to determine the post-transcriptional consequences of PRMT inhibition on RI (*Figure 3a*). Genome browser tracks demonstrated reduced RI and flanking exon signal intensity following 60 min of ActD treatment (*Figure 3b*). Further quantification of RI common to both GSK591- and MS023-treated cells relative to their non-ActD treated controls—normalized to the abundance of their flanking exons—demonstrated a negative median log$_2$ fold change in all conditions, indicating less RI abundance in the ActD-treated samples. Moreover, consistent with a post-transcriptional mechanism, we observed significantly decreased intron decay in GSK591-treated cells (median = −0.106) compared to DMSO (–0.220) or MS023-treated cells (–0.275) (p=0.002 relative to DMSO) (*Figure 3c*). The RI loss following ActD in MS023-treated cells was not significantly different than DMSO, although the median was more negative suggesting a trend toward increased decay (p=0.36) (*Figure 3c*).

We next validated these changes in RI following treatment with ActD using RT-qPCR for our three candidate introns described above (*Figure 1e*). GSK591 treatment resulted in significantly decreased intron decay relative to DMSO for *GAS5* intron 9 and *NOP2* intron 14 but not *ANKZF1* intron 9 after 60 min of ActD treatment (p=0.06, 0.0005, and 0.74, respectively) (*Figure 3d*). Conversely, intron decay increased with MS023 treatment relative to DMSO for *GAS5* intron 9 but not *ANKZF1* intron 9 or *NOP2* intron 14 (p=0.0043, 0.27, and 0.98, respectively) (*Figure 3d*). As the changes in RI following transcriptional inhibition with ActD reflect those of PRMT inhibition alone—increased RI with GSK591 and decreased RI with MS023—these results support the hypothesis that PRMTs regulate RI post-transcriptionally.

## PRMTs regulate the binding of RNA processing factors to chromatin-associated poly(A) RNA

We next sought to identify the factors responsible for mediating the post-transcriptional consequences of PRMT inhibition. Previous reports have highlighted that delayed co-transcriptional splicing leads to chromatin retention of poly(A) transcripts (*Brody et al., 2011*; *Pandya-Jones et al., 2013*; *Yeom et al., 2021*). Therefore, we analyzed differences in proteins bound to chromatin-associated

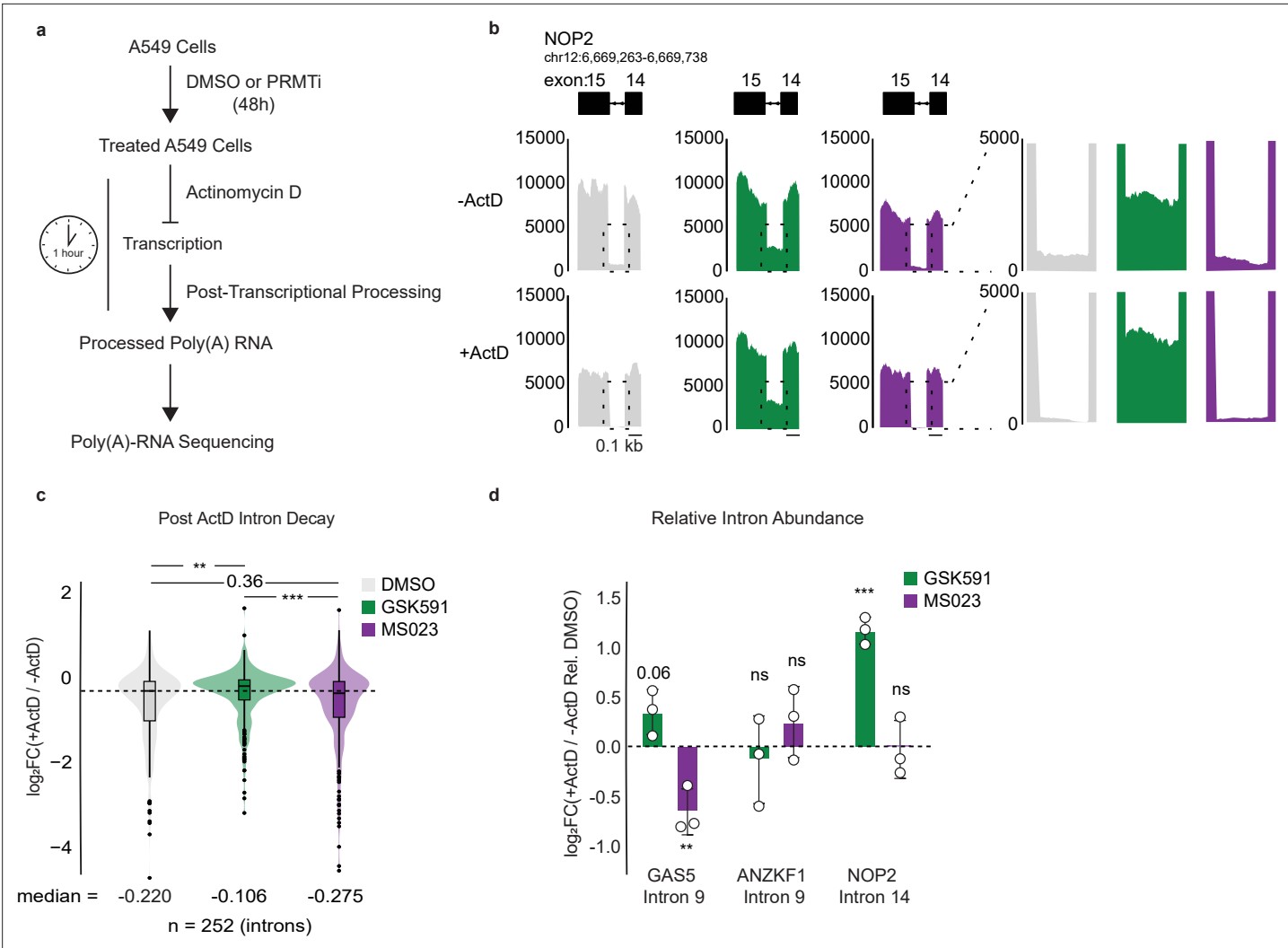

**Figure 3.** Type I and II PRMTs regulate RI post-transcriptionally. (**a**) Overview of Actinomycin D (ActD) poly(A)-RNA sequencing protocol. (**b**) Genome browser track of poly(A)-RNA seq aligned reads (−) and (+) ActD for *NOP2*. (**c**) Distribution of $\log_2$(+ActD/−ActD) abundance for common RI between GSK591 and MS023 treated cells relative to DMSO treatment. Dashed line indicates DMSO median; solid line within boxplot is condition-specific median. Significance determined using Wilcoxon rank-sum test; **<0.01, ***<0.001. (**d**) RT-qPCR of RI (−) and (+) ActD relative to DMSO. Data are represented as mean ± SD. Significance determined using Student's t-test; **<0.01, ***<0.001, ns=not significant.

poly(A) RNA. To accomplish this, we performed UV-crosslinking of A549 cells 2 days after treatment with DMSO, GSK591, or MS023. We then isolated the chromatin fraction, and after a high-salt wash, sheared the material with a light, brief sonication. This was followed by poly(A) enrichment using oligo(dT) beads. To control for non-specific interactions, we performed high stringency washes and added an excess of free poly(A) to our negative control. After elution of the bound poly(A) RNA, we digested the RNA and analyzed the remaining material with liquid chromatography coupled online with tandem mass spectrometry (LC-MS/MS) (***Figure 4a***).

Across the three conditions, we identified 1832 unique proteins on chromatin (***Supplementary file 1***). Following GSK591 treatment, 118 had significantly altered abundance with 70 increased and 48 decreased (p<0.05). In the MS023-treated chromatin, we observed 255 differentially enriched proteins, with 150 increasing in abundance and 105 decreasing (p<0.05). The chromatin-associated poly(A) enriched fraction—after background subtraction—contained 1251 unique proteins. Of these, 32 were differentially bound in the GSK591-treated samples with 24 increased and eight decreased (p<0.05). In the MS023-treated cells, there were 55 proteins with altered binding—37 increased and 18 decreased (p<0.05).

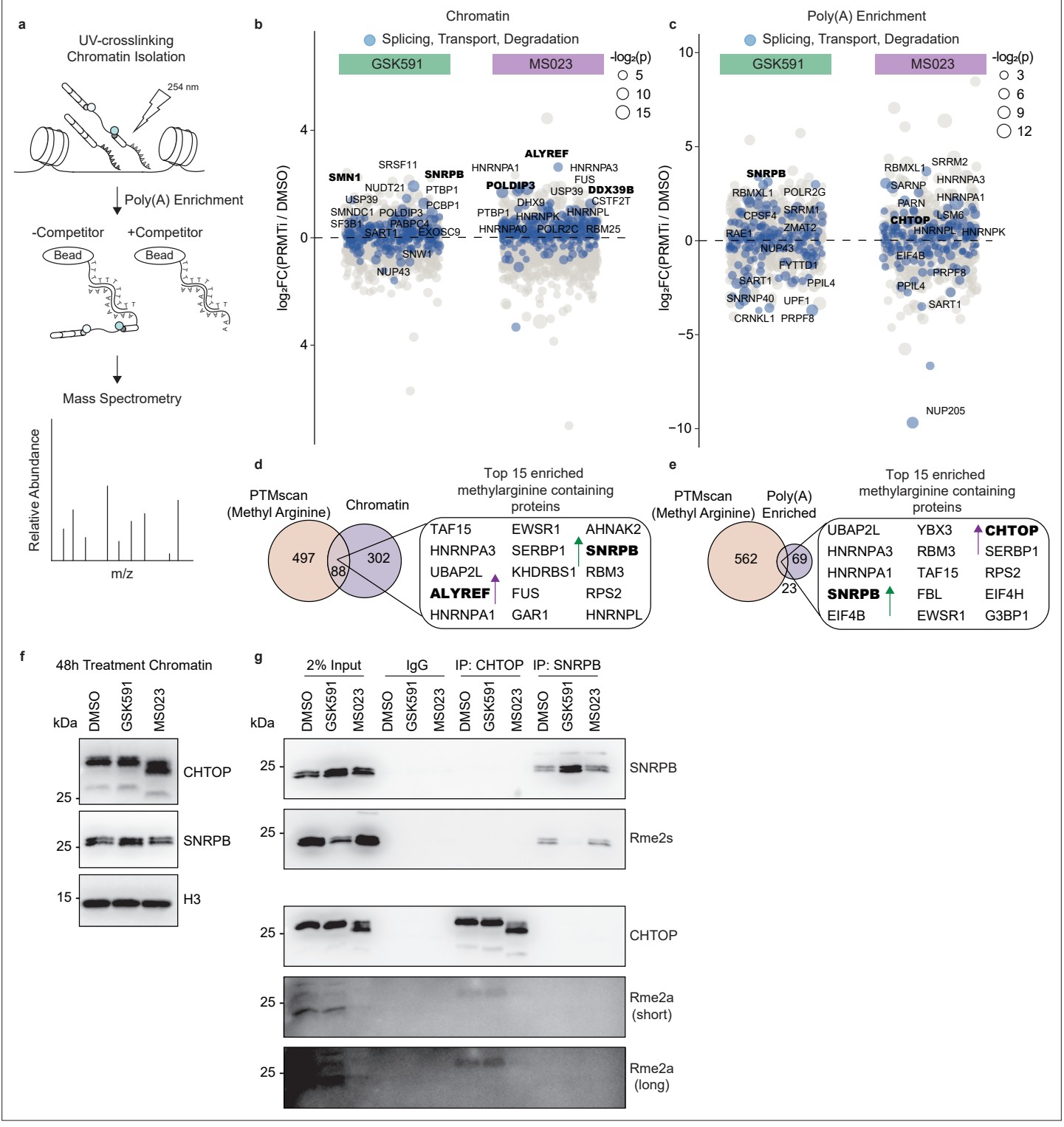

**Figure 4.** PRMT inhibition promotes aberrant binding of RNA processing factors to chromatin-associated poly(A) RNA. (**a**) Overview of chromatin-associated poly(A)-RNA LC-MS/MS experiment. (**b, c**) Dot plot of proteins bound to chromatin (**b**) or chromatin-associated poly(A) RNA (**c**) relative to DMSO. Circle size is proportional to −log₂(p). Colored values denote factors with ontology pertaining to RNA splicing, transport, or degradation. The names of the top 15 significant proteins are labeled. Significance determined using a heteroscedastic t-test. (**d, e**) Venn diagram comparing proteins containing methylarginine (*Maron et al., 2021*) and those that were differentially enriched in the chromatin (**d**) or chromatin-associated poly(A) (**e**) fractions following PRMT inhibition. (**f**) Western blot of chromatin following 2-day treatment with DMSO, GSK591, or MS023. See *Figure 4—source data 1*. (**g**) Immunoprecipitation and analysis of CHTOP and SNRPB methylarginine following treatment with DMSO, GSK591, or MS023 for 2 days. See

*Figure 4 continued on next page*

*Figure 4 continued*

**Figure 4—source data 2.**

The online version of this article includes the following source data and figure supplement(s) for figure 4:

**Source data 1.** Western blot data for *Figure 4f*.

**Source data 2.** Western blot data for *Figure 4g*.

**Figure supplement 1.** PRMT inhibition alters association of RNA splicing, localization, and processing factors to chromatin and chromatin-associated poly(A) RNA.

To identify the biological processes most affected by these inhibitors, we performed overrepresentation analysis of the top 200 enriched proteins (*Figure 4—figure supplement 1a and b*). In the chromatin fraction of both GSK591- and MS023-treated cells, we observed a significant enrichment of gene ontology terms for RNA splicing and ribonucleoprotein complex biogenesis (*Padj*<0.05) (*Figure 4—figure supplement 1a*). This was consistent with the gross aberrations in splicing following treatment with these inhibitors. In GSK591-treated cells, there was also a unique enrichment for the regulation of protein localization to Cajal bodies (*Padj*<0.05). Conversely, in MS023-treated cells, we observed a highly significant enrichment for nuclear RNA export and the regulation of RNA catabolism (*Padj*<0.05). When looking specifically at the poly(A)-enriched fraction, we noted similar ontologies as those in the chromatin fraction (*Figure 4—figure supplement 1b*).

## PRMT inhibition perturbs binding of nuclear export factors and snRNPs to poly(A) RNA

Of the proteins that were differentially bound to the input chromatin and poly(A) RNA following PRMT inhibition, we were interested in candidates that were most likely to mediate post-transcriptional RI. To accomplish this, we took into consideration the enriched gene ontology categories described above and highlighted the most significant factors involved in RNA splicing, transport, and degradation relative to their $\log_2$ fold change (*Figure 4b and c*). In the input chromatin of GSK591-treated cells, we observed increased signal in the snRNP assembly factor SMN, as well as the snRNP component SNRPB (SmB/B') (*Figure 4b*). We also observed enrichment of SNRPB in the poly(A)-enriched fraction (*Figure 4c*). Previous reports have documented changes in exon skipping following SNRPB knockdown—including within its own transcript—that resemble changes seen with PRMT5 knockdown (*Saltzman et al., 2011*; *Bezzi et al., 2013*). Together with the fundamental role of snRNPs in assembling the spliceosome, this suggested that SNRPB methylation may regulate intron retention.

In the input chromatin of MS023-treated cells, we observed an increase of the TREX components ALYREF, DDX39B, and POLDIP3 (*Figure 4b*). ALYREF and DDX39B are recruited to pre-mRNA co-transcriptionally and are crucial for the eventual hand-over of the mRNA to the nuclear export receptor, nuclear RNA export factor 1 (NXF1) (*Heath et al., 2016*). When looking specifically at the poly(A)-RNA fraction of MS023-treated cells, we noted an increase in CHTOP, also a component of the TREX complex (*Figure 4c*; *Heath et al., 2016*). Both CHTOP and ALYREF have been shown to be direct targets of the major Type I methyltransferase PRMT1 (*Hung et al., 2010*; *van Dijk et al., 2010*). In both cases, arginine methylation of ALYREF and CHTOP was demonstrated to weaken their binding to RNA and promote interaction with NXF1 (*Hung et al., 2010*; *Chang et al., 2013*). This is consistent with their increase in both the chromatin and poly(A)-RNA fractions following MS023-mediated inhibition of Type I PRMT activity.

To further refine the PRMT targets that regulate RI, we compared these data with the proteins that contained methylarginine identified in our recently published study of the A549 arginine methylome (*Maron et al., 2021*). In that work, we compiled our proteomic data with previously published arginine methylome data to establish a comprehensive database of methylarginine-containing proteins (*Maron et al., 2021*). When we intersected the arginine methylome with the differentially enriched proteins in our input and poly(A) enriched fractions (p<0.1), there were 88 and 23 unique methylarginine-containing proteins, respectively. We then analyzed the top 15 differentially enriched proteins in each fraction. In the input, we noted that ALYREF and SNRPB were both modified with methylarginine (*Figure 4d*). Furthermore, when looking at the poly(A) fraction, we again observed SNRPB as well as CHTOP (*Figure 4e*). The prior identification of ALYREF, CHTOP, and SNRPB as methylarginine-containing proteins suggested that these bonafide PRMT substrates may be potential mediators of

PRMT-dependent RI (*Brahms et al., 2001*; *Hung et al., 2010*; *van Dijk et al., 2010*; *Maron et al., 2021*).

To validate changes in CHTOP and SNRPB chromatin-association following PRMT inhibition, we isolated the chromatin fraction and probed for CHTOP and SNRPB. In MS023-treated cells, we observed faster CHTOP gel migration; this is consistent with its hypomethylation (*Figure 4f*). In GSK591-treated cells, there was increased chromatin-bound SNRPB (*Figure 4f*). To further confirm that the changes in methylarginine were specific to CHTOP and SNRPB, we performed an immunoprecipitation for these proteins following PRMT inhibition (*Figure 4g*). Subsequent analysis of methylarginine confirmed loss of CHTOP Rme2a with MS023 treatment and SNRPB Rme2s with GSK591 treatment (*Figure 4g*). Taken together, the differential enrichment of ALYREF, DDX39B, and POLDIP3 on chromatin, along with CHTOP on poly(A) RNA following treatment with MS023—in addition to their dependence on Rme2a—presented the possibility that the TREX complex may have an important role in the post-transcriptional consequences of Type I PRMTs on RI. Moreover, as snRNPs are fundamental in splicing, increased SNRPB in both the input and poly(A) fractions following treatment with GSK591—as well as the loss of Rme2s—prompted us to further evaluate the role of SNRPB in the regulation of RI.

## Knockdown of CHTOP and SNRPB recapitulates changes in RI seen following PRMT inhibition

To address the question of whether CHTOP, ALYREF, or SNRPB were involved in regulation of RI, we first checked whether there was any publicly available RNA-seq data in which these proteins were perturbed. We found two independent data sets where SNRPB was knocked down in U251 glioblastoma cells or in HeLa cells (*Saltzman et al., 2011*; *Correa et al., 2016*). We also identified two independent data sets where CHTOP or ALYREF were knocked down in HEK293T cells or HeLa cells, respectively (*Fan et al., 2019*; *Viphakone et al., 2019*). Strikingly, after performing rMATS on these data sets, we observed that both SNRPB knockdown (SNRPBkd) experiments strongly recapitulated the increase in RI seen with GSK591 treatment (*Figure 5a*). Likewise, although ALYREF knockdown did not significantly affect RI levels (*Figure 5—figure supplement 1a*), CHTOP knockdown (CHTOPkd) resulted in a global decrease in RI inclusion, paralleling that seen with MS023 treatment (*Figure 5a*). To understand if the RI were common across data sets, we intersected either the SNRPBkd or CHTOPkd RI with the RI seen following GSK591 or MS023 treatment (*Figure 5b and c*). There were 535 mutual RI between GSK591 and both the SNRPBkd data sets, most of which had the same effect on $\Delta \Psi$ ($\log_2$ OR 7.84 for GBM and 7.56 for HeLa, p<1e−300) (*Figure 5b*). The opposite was true when comparing GSK591 to CHTOPkd—we observed 620 shared RI ($\log_2$ OR 7.49, p<1e−300) that largely contrasted in their $\Delta \Psi$ (*Figure 5b*). When compared to SNRPBkd and CHTOPkd, MS023 treatment had the inverse effect to GSK591 treatment. There were 164 overlapping RI between both SNRPBkd data sets ($\log_2$ OR 6.68 for GBM and 6.38 for HeLa, p<7e−290) with the majority having opposite $\Delta \Psi$ values; CHTOPkd and MS023 treatment shared 201 RI ($\log_2$ OR 6.46, p<2e−279), with the majority having coincident $\Delta \Psi$ values (*Figure 5c*). Consistent with PRMTs regulating a common group of RI, SNRPBkd and CHTOPkd were significantly correlated with either GSK591 or MS023 treatment (*Figure 5d*). Remarkably, the correlations that were positive for GSK591 treatment were negative for MS023 treatment, reflecting the changes seen in poly(A)-RNA seq and consistent with SNRPB and CHTOP mediating the effects of PRMT inhibition on RI.

## Sm arginine mutants increase RI

To address the question of whether Sm arginine methylation was directly involved in the regulation of RI, we designed a single vector containing SNRPD3, SNRPB, and SNRPD1—the Sm targets of PRMT5—as either wild-type (WT), R-to-A, or R-to-K mutants separated by 2a-self cleaving peptides (P2A) and modified with FLAG- (SNRPD3 and SNRPB) or V5-affinity (SNRPD1) tags at their N-termini (*Figure 5e*). To identify the appropriate arginines to mutate, we utilized our proteomic data (*Maron et al., 2021*) as well as a previously published high-resolution crystal structure of the U4/U6.U5 tri-snRNP (*Charenton et al., 2019*). As we showed that PRMTs prefer to methylate intrinsically disordered regions (IDRs), we mutated all 29 arginines within Sm IDRs and within GR repeats (*Figure 5e*; *Maron et al., 2021*). We then transduced A549 cells with the constructs. We achieved similar expression of the exogenous Sm proteins relative to the endogenous between WT and mutant constructs at the level of RNA (*Figure 5—figure supplement 1b*). However, likely owing to the tight cellular control

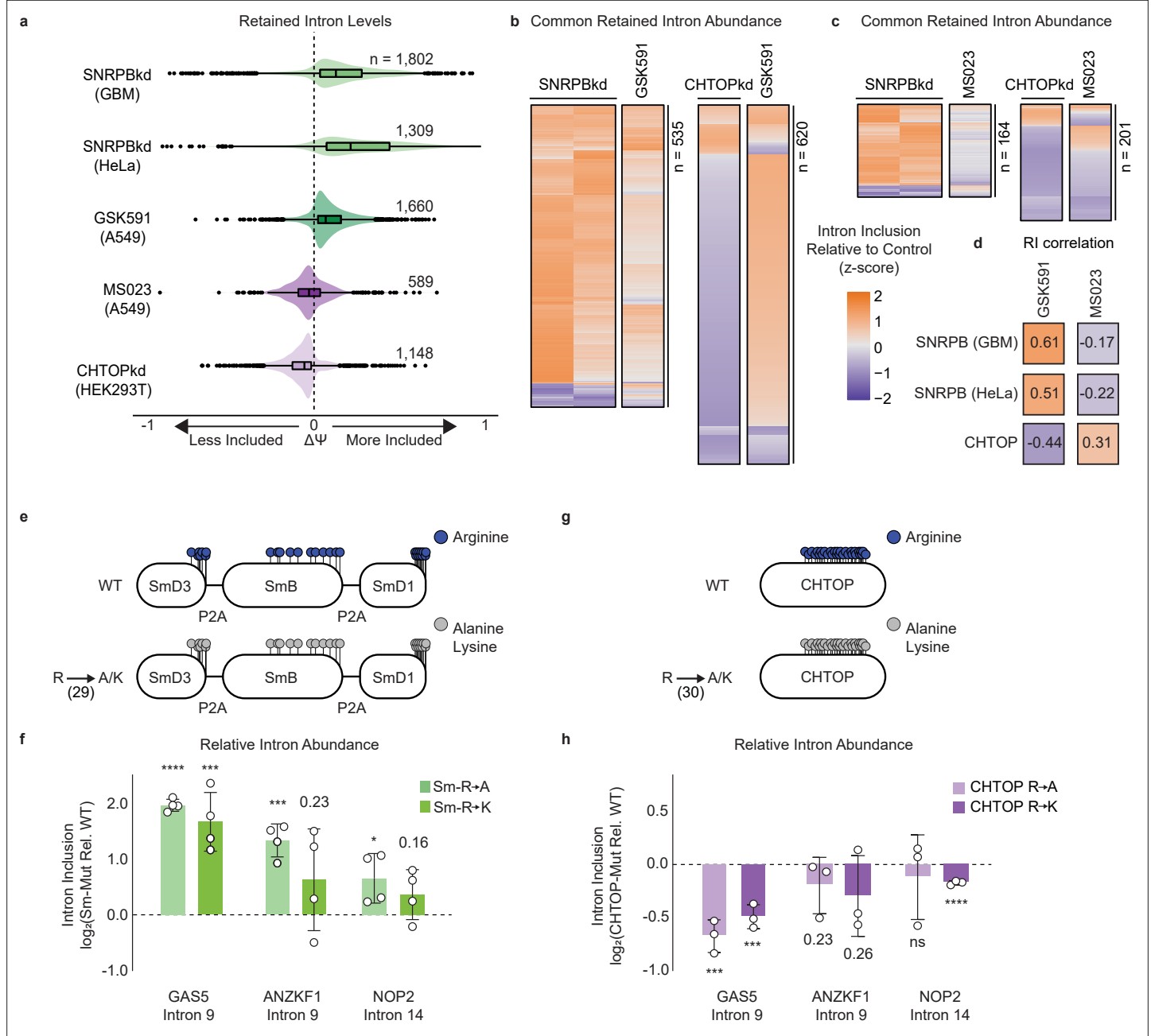

**Figure 5.** Sm and CHTOP methylarginine mediates PRMT-dependent RI. (**a**) Comparison of $\Delta\Psi$ for RI following SNRPB knockdown (kd), GSK591 treatment, MS023 treatment, and CHTOPkd where $\Delta\Psi = \Psi$ (Treatment)– $\Psi$ (Control). (**b, c**) Comparison of $\Delta\Psi$ z-score for common RI between SNRPBkd and CHTOPkd with either GSK591 (**b**) or MS023 (**c**) treatment. (**d**) Spearman rank correlation of $\Delta\Psi$ for common RI between SNRPBkd and CHTOPkd with either GSK591 or MS023 treatment. (**e**) Schematic of Sm expression constructs where lollipops represent individual methylarginines. (**f**) RT-qPCR of RI following transduction with Sm R-to-A or R-to-K mutants relative to wild-type (WT). Data are represented as mean ± SD. Significance determined using Student's t-test; *<0.05, ***<0.001, ****<0.0001. (**g**) Schematic of CHTOP expression construct as in panel (**e**). (**h**) RT-qPCR of RI following transduction with CHTOP R-to-A or R-to-K mutants relative to WT. Data are represented as mean ± SD. Significance determined using Student's t-test; ***<0.001, ****<0.0001, ns=not significant.

The online version of this article includes the following source data and figure supplement(s) for figure 5:

**Figure supplement 1.** CHTOP and Sm arginine mutants phenocopy PRMT inhibition.

**Figure supplement 1—source data 1.** Western blot data for *Figure 5—figure supplement 1c*.

**Figure supplement 2.** Sm methylarginine is dispensable for snRNP assembly.

**Figure supplement 2—source data 1.** Western blot data for *Figure 5—figure supplement 2a*.

*Figure 5 continued on next page*

*Figure 5 continued*

**Figure supplement 2—source data 2.** Northern blot data for *Figure 5—figure supplement 2b*.

**Figure supplement 2—source data 3.** Northern blot data for *Figure 5—figure supplement 2c*.

**Figure supplement 2—source data 4.** Image data for *Figure 5—figure supplement 2d*.

**Figure supplement 2—source data 5.** Image data for *Figure 5—figure supplement 2e*.

of total Sm levels, we were unable to achieve ample overexpression with the WT vector (*Figure 5— figure supplement 1c*; *Prusty et al., 2017*). We observed a migratory shift toward a lower molecular weight in the R-to-A mutants when compared to the WT or R-to-K mutants (*Figure 5—figure supplement 1c*). We also observed a strong Rme2s signal on the FLAG-SmB WT protein that was absent on the R-to-A and R-to-K mutants (*Figure 5—figure supplement 1c*).

As the Sm proteins exist in a heptameric ring bound to snRNA—together forming snRNPs—we tested whether mutating the arginines of the C-terminal tail compromised snRNP assembly. To accomplish this, we transduced A549 cells with the Sm WT, R-to-A, or R-to-K mutants and performed a co-immunoprecipitation targeting the N-terminal FLAG tag. We then used western and northern blotting to probe for snRNP-associated proteins and snRNAs, respectively. When analyzing binding to the snRNP assembly factor, SMN, we did not observe any differences between the WT, R-to-A, or R-to-K constructs (*Figure 5—figure supplement 2a*). Furthermore, we enriched the U1 snRNP specific factor, U1-70k, and confirmed a complete absence of Rme2s with R-to-A and R-to-K mutagenesis (*Figure 5—figure supplement 2a*). The ability of the Sm mutants to bind snRNA was also unperturbed: we observed an enrichment of all the major snRNAs—U1, U2, U4, U5, and U6—with the WT, R-to-A, and R-to-K constructs (*Figure 5—figure supplement 2b*). To confirm that the total cellular pool of snRNPs was not compromised, we also immunoprecipitated SNRPB; this should enrich both endogenous and exogenous Sm proteins (*Figure 5—figure supplement 2b*). We did not observe any differences in snRNA binding to SNRPB. This supports the conclusion that cellular snRNP assembly is not dependent on methylarginine or compromised by the expression of the exogenous Sm proteins. Furthermore, and consistent with the dispensability of Sm methylarginine for snRNP assembly, extensive treatment of A549 cells with GSK591 similarly did not disrupt binding of Sm proteins to snRNA (*Figure 5—figure supplement 2c*). Importantly, these results are in line with prior published reports where deletion of *dart5*—the *Drosophila* PRMT5 orthologue—or deletion of the Sm tails did not affect binding of Sm proteins to SMN and snRNP assembly (*Gonsalvez et al., 2006*; *Chari et al., 2008*).

In addition to accounting for snRNP assembly, we also tested whether the mutagenesis of the Sm C-terminal domain altered snRNP nuclear import by performing indirect immunofluorescence for either FLAG or SNRPB following transduction of the Sm mutants (*Figure 5—figure supplement 2d*). Using this approach, we observed both cytoplasmic and nuclear signal with FLAG and SNRPB antibodies in the WT, R-to-A, and R-to-K Sm-expressing cells (*Figure 5—figure supplement 2d*). We performed a similar experiment using indirect immunofluorescence for SNRPB following treatment with DMSO, GSK591, or MS023 and similarly observed both cytoplasmic and nuclear signal (*Figure 5—figure supplement 2e*). Taken together, these data support that arginine methylation of Sm proteins is dispensable for both snRNP assembly and nuclear import.

As our primary question was how Sm methylarginine influences RI levels, we analyzed intron inclusion in our three candidate transcripts, *GAS5* intron 9, *ANKZF1* intron 9, and *NOP2* intron 14 following transduction with Sm WT, R-to-A, or R-to-K mutants. With the R-to-A Sm mutants, we observed increased inclusion of all three RI ($p<0.05$) (*Figure 5f*). We also detected a significant increase in *GAS5* intron 9 ($p<0.001$) with the Sm R-to-K mutants and a trend toward increased RI in *ANKZF1* intron 9 ($p=0.23$) and *NOP2* intron 14 ($p=0.16$). Thus, mutagenesis of Sm methylarginine sites increased RI in our candidate transcripts. The greater effect of the R-to-A mutants suggests that the charge of arginine itself may also play an important role in regulating RI levels.

## CHTOP arginine mutants decrease RI

We performed similar experiments with CHTOP, in which we mutated 30 arginines present within the centrally located 'GAR' motif—the preferred PRMT1 substrate recognition motif—to either alanine or lysine (*Figure 5g*). This region of CHTOP has been previously shown to be required for PRMT1-catalyzed methylarginine (*van Dijk et al., 2010*). We transduced A549 cells with the constructs and

performed RT-qPCR for *GAS5* intron 9, *ANKZF1* intron 9, or *NOP2* intron 14. We achieved similar expression of the mutant CHTOP proteins when compared to the WT at the level of RNA and protein (*Figure 5—figure supplement 1c and d*). Interestingly, when transducing cells with the WT CHTOP, we noted a gross downregulation of the endogenous transcript (*Figure 5—figure supplement 1e*). CHTOP has been previously reported to control its own expression as part of an autoregulatory loop (*Izumikawa et al., 2016*). We did not observe this compensation with either the R-to-A or R-to-K mutants (*Figure 5—figure supplement 1e*). Consistent with the gel shift in CHTOP seen following MS023 treatment, we observed a similar change in the R-to-A mutant when compared to the WT or R-to-K mutants. Moreover, supporting a role for CHTOP methylarginine in Type I PRMT-dependent RI inclusion, in both R-to-A and R-to-K mutants, we observed decreased inclusion in *GAS5* intron 9 (p<0.001) (*Figure 5h*). We also saw a trend toward decreased inclusion of *ANKZF1* intron 9 (p=0.23 and 0.26) and significantly decreased inclusion of *NOP2* intron 14 (p<0.0001) in the R-to-K mutant, but not the R-to-A mutant. Taken together, these experiments support that CHTOP methylarginine is involved in regulating RI levels.

## PRMT-regulated RI are detained within the nucleoplasm and chromatin

PRMT5 has been proposed to specifically regulate DI—introns that persist in poly(A) RNA but remain nuclear (*Braun et al., 2017*). However, direct experimental evidence—namely fractionation of subcellular compartments to identify the location of RI—has been lacking. To address the question of whether the RI in our data are nuclear and therefore DI, we first ran rMATS on publicly available ENCODE poly(A)-RNA seq from cytoplasmic and nuclear fractions of A549 cells (ENCSR000CTL and ENCSR000CTM, respectively). We observed an enrichment of RI within the nuclear fraction: 97% of significant RI events (FDR<0.05) had a $+\Delta\Psi$, where $\Delta\Psi$ is the difference in $\Psi$ between the nuclear and cytoplasmic compartments. We then intersected these data with the PRMT-dependent RI and noted that there was a significant overlap with both GSK591 ($\log_2$ OR 8.89, p<1e−300) and MS023 treatment ($\log_2$ OR 7.75, p<1e−300) when compared to all A549 expressed introns (*Figure 6a*). We also observed that *GAS5* intron 9, *ANKZF1* intron 9, and *NOP2* Intron 14 were significantly increased in nuclear poly(A) RNA (*GAS5* and *ANKZF1* not shown) (*Figure 6b*).

To validate that these RI remain nuclear following PRMT inhibition, and to further resolve their localization within the nucleus, we treated A549 cells with DMSO, GSK591, and MS023 and isolated cytoplasmic, nucleoplasmic, and chromatin fractions. We first confirmed the integrity of our fractionation approach by comparing the distribution of GAPDH, LaminB2, U1-70k, and H3 (*Figure 6c*). Whereas GAPDH was strongly enriched in the cytoplasm, LaminB2 was completely nuclear and distributed between the chromatin and nucleoplasm. Furthermore, U1-70k was nucleoplasmic, while H3 was only found on chromatin. This is consistent with results from previously published reports (*Bhatt et al., 2012*; *Yeom et al., 2021*). We next analyzed the cellular distribution of SNRPB, SNRPD3, and CHTOP as well as Rme2s and Rme2a. Interestingly, we observed an increase in nucleoplasmic and chromatin-bound SNRPB in cells treated with GSK591, with the majority of SNRPB located in the nucleoplasm (*Figure 6c*). A similar distribution was also seen for SNRPD3. When analyzing Rme2s across cellular fractions, there was a significant decrease in the cytoplasmic, nucleoplasmic, and chromatin fractions with GSK591 treatment and a slight increase in Rme2s signal with MS023 treatment that was most apparent in the nucleoplasm (*Figure 6c*). When analyzing the distribution of CHTOP across cellular fractions, we observed an enrichment in the nucleoplasm and chromatin; a migration shift toward a lower molecular weight with MS023 treatment was also seen in both fractions (*Figure 6c*). Furthermore, MS023 resulted in a strong decrease in Rme2a across all cellular fractions, while GSK591 treatment slightly increased Rme2a levels in the nucleoplasm and chromatin (*Figure 6c*).

We then quantified levels of *GAS5* intron 9, *ANKZF1* intron 9, *NOP2* intron 14, and *HNRNPH1* intron 14 or non-RI within the same poly(A) transcripts in the cytoplasmic, nucleoplasmic, and chromatin fractions (*Figure 6d*). In accordance with the RI being localized to the nucleus, in all conditions we observed a significant increase in nucleoplasmic and chromatin RI signal compared to the cytoplasm (p<0.05) (*Figure 6d*). Importantly, for the non-RI, there was no significant difference between cytoplasmic or nuclear enrichment. Taken together, these data support that these RI are indeed localized to the nucleoplasm and chromatin fractions and are thus detained introns.

We next wanted to understand which cellular fraction had the greatest changes in post-transcriptional intron decay following PRMT inhibition. To accomplish this, we pre-treated cells with DMSO, GSK591,

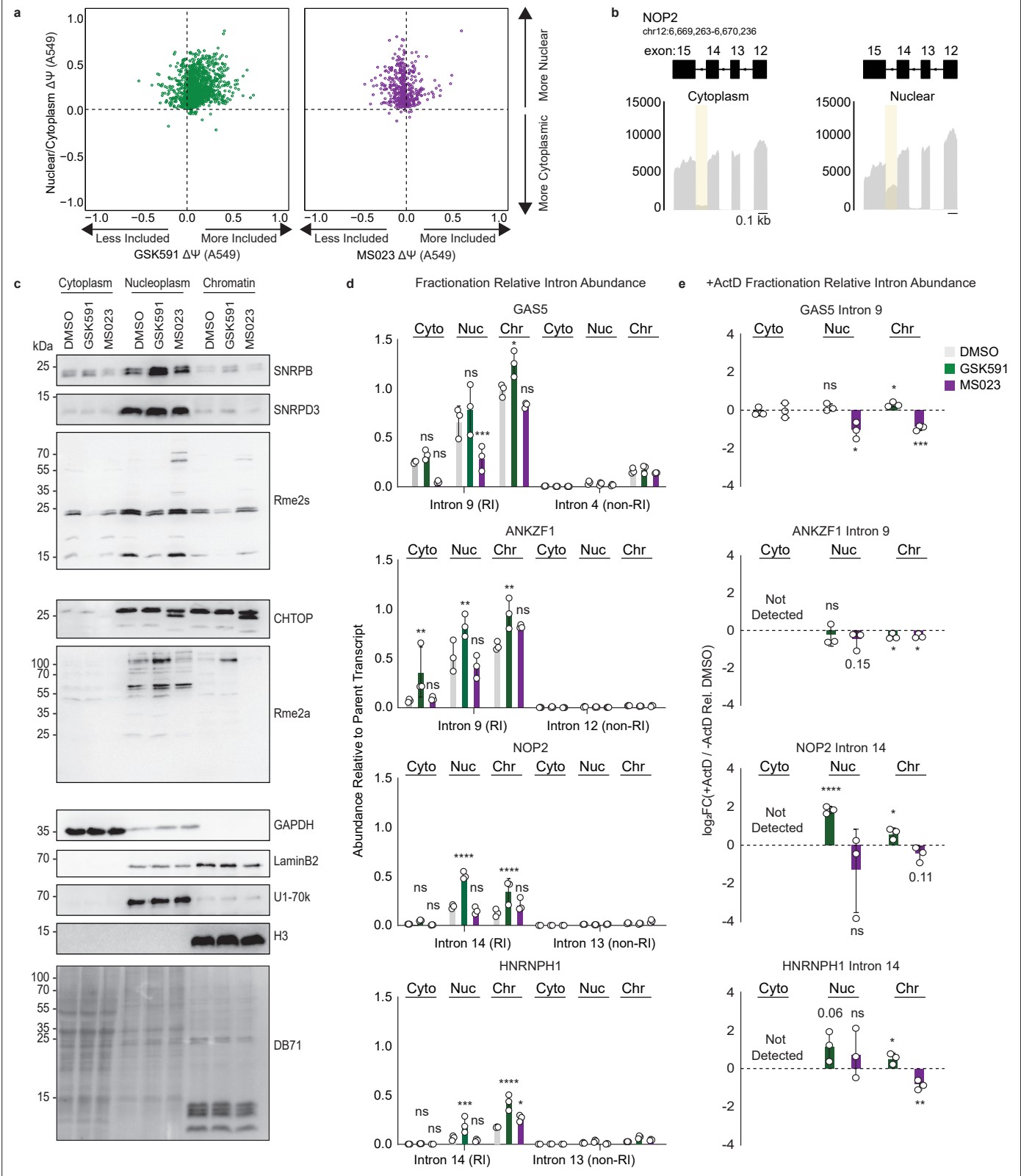

**Figure 6.** PRMT-dependent RI are localized to the nucleoplasm and chromatin. (**a**) Scatter plot of $\Delta\Psi$ for common RI in A549 nuclear/cytoplasmic fractions and GSK591 (left, green) or MS023 (right, purple) treated cells where $\Delta\Psi = \Psi$ (Nuclear/PRMT inhibition)– $\Psi$ (Cytoplasm/DMSO). (**b**) Genome browser track of poly(A)-RNA seq aligned reads for NOP2 in A549 cytoplasmic or nuclear fractions. (**c**) Western blot of cellular fractions following 2-day treatment with DMSO, GSK591, or MS023. DB71=Direct Blue 71 membrane stain. See ***Figure 6—source data 1***. (**d**) RT-qPCR of RI or non-RI

*Figure 6 continued on next page*

*Figure 6 continued*

from cytoplasmic, nuclear, or chromatin fractions of A549 cells treated with DMSO, GSK591, or MS023 for 2 days. Data are represented as mean ± SD. Significance determined using two-way analysis of variance with Tukey's multiple comparisons test; *<0.05, **<0.01, ***<0.001, ****<0.0001, ns=not significant. (**e**) RT-qPCR of RI (−) and (+) ActD relative to DMSO from cytoplasmic, nuclear, or chromatin fractions. Data are represented as mean ± SD. Significance determined using Student's t-test; *<0.05, **<0.01, ***<0.001, ****<0.0001, ns=not significant.

The online version of this article includes the following source data for figure 6:

**Source data 1.** Western blot data for *Figure 6c*.

or MS023 for 2 days and then blocked transcription with ActD for 60 min (*Figure 3a*). We then fractionated the cells into cytoplasmic, nucleoplasmic, and chromatin components and analyzed RI levels. When compared to DMSO treatment, we observed that the changes in intron decay with GSK591 or MS023 treatment were isolated to the nucleoplasmic and chromatin fractions (*Figure 6e*). Specifically, GSK591 treatment resulted in decreased chromatin-associated intron decay in *GAS5* intron 9, *NOP2* intron 14, and *HNRNPH1* intron 14 (p<0.05), while a slight increase was observed for *ANKZF1* intron 9 (p<0.05) (*Figure 6e*). In the nucleoplasm, *NOP2* intron 12 (p<0.0001) and *HNRNPH1* intron 14 (p=0.06) also demonstrated less intron decay, while there was no significant difference in *GAS5* intron 9 and *ANKZF1* intron 9 (p>0.05) (*Figure 6e*). MS023 treatment promoted increased intron decay in chromatin-associated *GAS5* intron 9 (p<0.001), *ANKZF1* intron 9 (p<0.05), *NOP2* intron 14 (p=0.11), and *HNRNPH1* intron 14 (p<0.01) (*Figure 6e*). These changes were less pronounced in the nucleoplasm, where only *GAS5* intron 9 (p<0.05) had significantly greater decay relative to DMSO treatment (*Figure 6e*). Taken together, these results recapitulate the changes in RI decay seen with whole-cell ActD poly(A) RNA sequencing—decreased intron decay with GSK591 treatment and increased decay with MS023 treatment—with enhanced resolution of the cellular compartments in which they occur: the nucleoplasm and chromatin.

In summary, we conclude that Type I and II PRMTs regulate post-transcriptional DI through modulation of snRNP and CHTOP arginine methylation (*Figure 7a*). We also propose the following model for PRMT-dependent DI regulation: (1) snRNP Rme2s is required for post-transcriptional maturation of DI-containing transcripts—in the absence of Rme2s these snRNPs are bound to transcript but the DI are sequestered from splicing or degradation (*Figure 7b*), (2) CHTOP Rme2a is necessary for DI-containing transcripts to progress through the TREX-mediated nuclear export pathway; CHTOP lacking Rme2a inhibits this progression but does not protect the poly(A) RNA, which together with the consequent increase in nuclear residence time promotes splicing or degradation (*Figure 7c*).

## Discussion

In this study, our goal was to characterize the consequences of PRMT inhibition on RI. Many reports have documented that Type II PRMTs have a crucial role in splicing (*Boisvert et al., 2002*; *Bezzi et al., 2013*; *Koh et al., 2015*; *Yang et al., 2015*; *Braun et al., 2017*; *Fedoriw et al., 2019*; *Fong et al., 2019*; *Radzisheuskaya et al., 2019*; *Tan et al., 2019*; *Li et al., 2021*; *Sachamitr et al., 2021*). However, the role of Type I PRMTs in splicing has only recently been appreciated (*Fedoriw et al., 2019*; *Fong et al., 2019*). This is the first study to report that Type I PRMT inhibition decreases DI. Interestingly, simultaneous Type I and II PRMT inhibition increased DI (*Figure 1b*), suggesting that Type II catalyzed Rme2s is dominant in DI regulation. This observation is consistent with our model, whereby snRNP binding occurs upstream of CHTOP-mediated intron detention and protects RNA from post-transcriptional splicing or degradation (*Figure 7a*).

Whether PRMTs regulate splicing co- or post-transcriptionally has not previously been explored. We found that Type I PRMT inhibition promoted slower co-transcriptional splicing, while paradoxically resulting in fewer DI. Given that arginine methylation has been shown to disrupt protein-RNA binding (*Blackwell and Ceman, 2012*), one possible mechanism for this observation may be that the absence of Rme2a inhibits or delays the dissociation of RNA processing factors from nascent RNA thereby leading to slower splicing. We and others found that PRMT-dependent DI share characteristics such as non-canonical splice sites, shorter length, and location closer to the TES that makes their co-transcriptional removal less likely (*Wong et al., 2013*; *Braunschweig et al., 2014*; *Boutz et al., 2015*; *Braun et al., 2017*). As has been proposed by others, DI are likely an evolutionarily conserved class of introns used for fine tuning the transcriptome (*Braunschweig et al., 2014*; *Boutz et al.,*

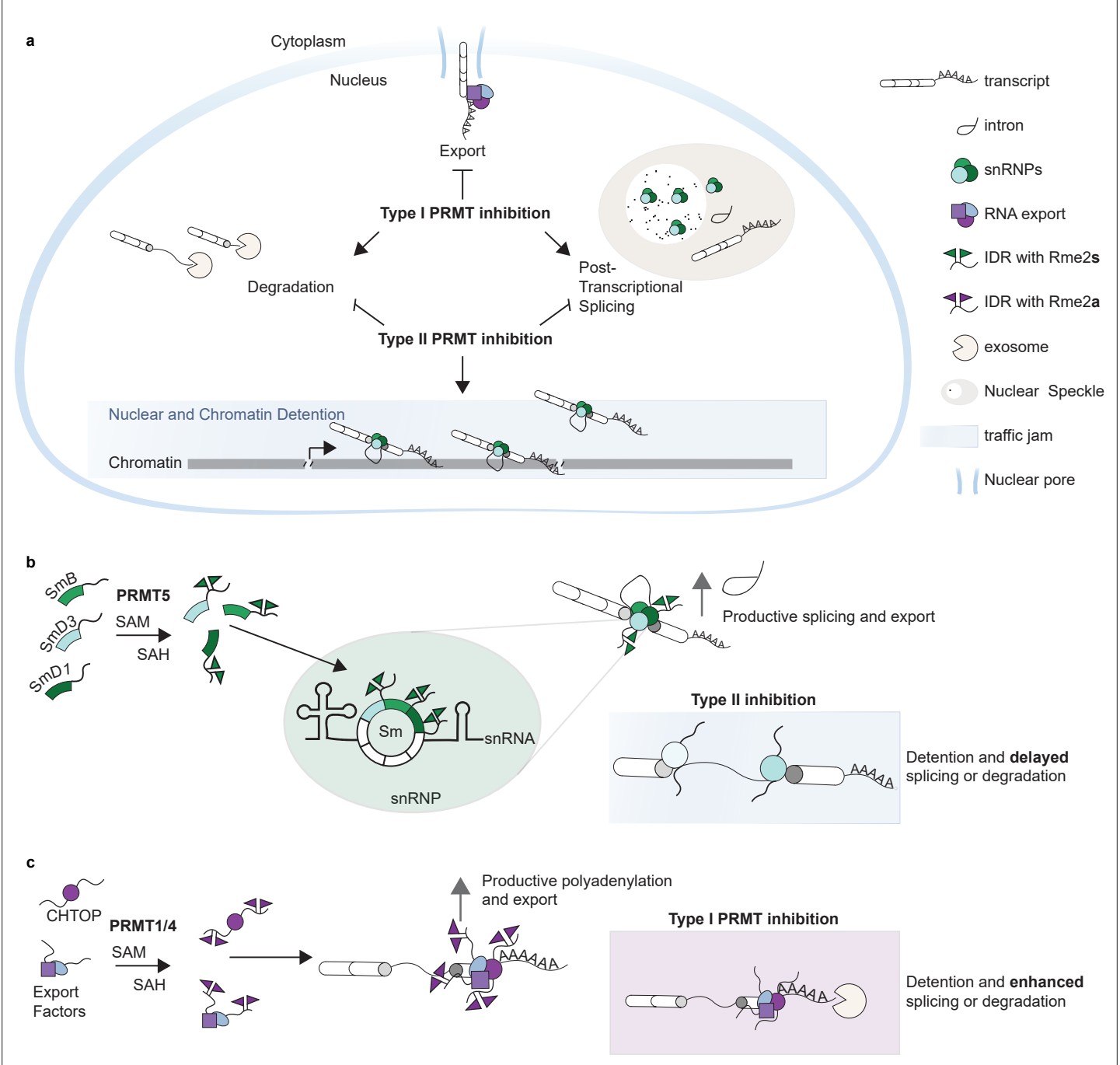

**Figure 7.** Type I and II PRMTs regulate post-transcriptional intron detention through Sm and CHTOP methylation. (**a–c**) Model figures representing an overview of Type I- and Type II-dependent regulation of detained introns (DI) (**a**). Sm Rme2s is necessary for productive splicing and maturation of DI; absence of Rme2s results in detention of the parent transcript (**b**). Type I PRMT-catalyzed CHTOP Rme2a is required for appropriate polyadenylation and nuclear export of DI; absence of Rme2a increases nuclear residence time resulting in increased degradation or splicing (**c**).

*2015*; *Pimentel et al., 2016*). This is further supported by our comparison of DI in publicly available data sets, which—despite using diverse cell models and methods of inhibiting PRMTs—had a strong overlap with the DI present in our data. The same characteristics that lead these introns to become detained are likely what make them more sensitive to perturbation of splicing components. Consistent with this model, DI had slower splicing rates relative to non-DI and were transcribed significantly faster. Together with their TES-proximal location, this combination is likely to decrease the

window of opportunity for splicing to be completed co-transcriptionally and increase their reliance on post-transcriptional processing. Indeed, when using actinomycin D to study post-transcriptional DI processing, Type II PRMT inhibition resulted in decreased DI loss compared to DMSO or Type I PRMT inhibition, whereas Type I PRMT inhibition increased DI loss relative to DMSO and Type II PRMT inhibition. Thus, the post-transcriptional—rather than co-transcriptional—consequences reflect those seen in poly(A) RNA.

Although little is known about how post-transcriptional splicing is accomplished, previous work has demonstrated that post-transcriptional spliceosomes are retained in nuclear speckles (*Girard et al., 2012*). Furthermore, CHTOP and the TREX complex have also been found to localize within nuclear speckles (*Dias et al., 2010*; *Chang et al., 2013*). Perturbation of either splicing factors or TREX components resulted in accumulation of poly(A) RNA in nuclear speckles (*Dias et al., 2010*; *Chang et al., 2013*). It is therefore possible that the two types of PRMTs coalesce at the level of this membraneless organelle. Consistently, we observed that SRRM1 was enriched in the poly(A) fraction in Type II inhibited cells, whereas SRRM2 was enriched in the poly(A) fraction in Type I inhibited cells (*Figure 5b*). Many SR-repeat proteins—including SRRM1 and SRRM2—are localized to nuclear speckles (*Ilik et al., 2020*). Given the importance of arginine methylation in regulating liquid-liquid phase separation (LLPS) and the role of LLPS in maintaining nuclear speckle integrity—as well as other membraneless organelles such as Cajal bodies—PRMT inhibition may result in failed coordination and spatial localization of RNA processing factors leading to aberrant post-transcriptional processing (*Courchaine et al., 2021*; *Liao and Regev, 2021*).

As delayed splicing has been shown to result in chromatin retention of transcripts (*Brody et al., 2011*; *Pandya-Jones et al., 2013*; *Yeom et al., 2021*), we assayed chromatin-associated poly(A) RNA to identify factors that regulate PRMT-dependent DI. By interrogating changes in proteins bound to chromatin-associated poly(A) RNA—and their methylarginine levels—we identified snRNPs as key-mediators of DI regulation. Sm arginine methylation occurs on their intrinsically disordered C-terminal tail domains (*Brahms et al., 2000*; *Brahms et al., 2001*). Consistently, mutagenesis of arginine within the Sm C-terminal tails to either alanine or lysine abrogated Rme2s. Interestingly, loss of Sm Rme2s did not affect snRNP assembly. As the Sm-protein tails have previously been shown to be dispensable for snRNP assembly, and knockout of the *Drosophila* PRMT5 orthologue had no effect on snRNP biogenesis, our data strongly suggest that the Sm C-terminal tails and their modification with methylarginine are important for post-transcriptional RNA processing (*Gonsalvez et al., 2006*; *Chari et al., 2008*). Furthermore, because the Sm C-terminal domains are intrinsically disordered and intrinsically disordered proteins are known to be heavily involved in LLPS, it is possible that methylarginine on the Sm tails regulates incorporation into membraneless organelles required for post-transcriptional processing (*Courchaine et al., 2021*).

When analyzing proteins differentially bound to chromatin-associated poly(A) RNA following PRMT inhibition, we also identified the TREX complex component, CHTOP. Importantly, CHTOP contains a RI within its own transcript that is used to autoregulate its expression by nonsense-mediated decay (*Izumikawa et al., 2016*). This is accomplished by antagonizing intron excision by HNRNPH when CHTOP—using its 'GAR'-motif—binds to its own transcript (*Izumikawa et al., 2016*). Therefore, it is possible that a similar mechanism may be occurring globally whereby introns that are slow to splice are bound by CHTOP—which has been shown to bind nascent RNA co-transcriptionally—and this prevents both further binding by HNRNP proteins and intron excision (*Izumikawa et al., 2016*; *Viphakone et al., 2019*).

Another mechanism may be explained by the role of CHTOP in alternative polyadenylation (APA) (*Viphakone et al., 2019*). Previous work demonstrated that CHTOP specifically binds to the 3′ end of transcripts and CHTOP knockdown promotes changes in APA (*Viphakone et al., 2019*). The length of poly(A) is a driver of exosome-mediated decay, whereby abnormally long poly(A) tails—seen in DI-containing transcripts—increases exosome-mediated decay (*Bresson and Conrad, 2013*; *Bresson et al., 2015*; *Muniz et al., 2015*). Hyperadenylation occurs through poly(A) binding protein nuclear 1 (PABPN1) dependent stimulation of poly(A) polymerases (*Bresson and Conrad, 2013*). In MS023-treated cells, PABPN1 was specifically enriched in the chromatin and poly(A) fraction (p<0.1) (*Supplementary file 1*). Thus, loss of CHTOP Rme2a may promote hyperadenylation and consequent exosomal degradation of DI-containing transcripts. Future studies will probe the role of the exosome in PRMT-regulated protection or decay of DI.

As PRMTs have thousands of diverse substrates—many of which are involved in RNA homeostasis—the effect of PRMTs on splicing integrity is likely to be multifactorial (*Guccione and Richard, 2019*; *Lorton and Shechter, 2019*). Here, we highlighted both snRNPs and CHTOP as key mediators of PRMT-regulated DI. However, given the importance of both CHTOP and snRNPs in global RNA processing, why only a subset of alternative splicing events and transcripts are affected by their perturbation remains to be determined. Importantly, there are likely many other factors involved in regulating PRMT-dependent alternative splicing. Future work will be needed to characterize the methylarginine dependent mechanisms of post-transcriptional RNA processing.

# Materials and methods

**Key resources table**

| Reagent type (species) or resource | Designation | Source or reference | Identifiers | Additional information |
|---|---|---|---|---|
| Cell line (*Homo sapiens*) | A549 | ATCC | Cat#: CCL-185 RRID:CVCL_0023 | |
| Transfected construct (*H. sapiens*) | Wild-type FLAG-SmD3-p2a-FLAG-SmB-p2a-V5-SmD1 | VectorBuilder | VB210103-1026dg | Lentiviral construct to transfect and express the wild-type Sm proteins. |
| Transfected construct (*H. sapiens*) | R-to-A FLAG-SmD3-p2a-FLAG-SmB-p2a-V5-SmD1 | VectorBuilder | VB210103-1027ttz | Lentiviral construct to transfect and express the R-to-A mutant Sm proteins. |
| Transfected construct (*H. sapiens*) | R-to-K FLAG-SmD3-p2a-FLAG-SmB-p2a-V5-SmD1 | VectorBuilder | VB210317-1185yxr | Lentiviral construct to transfect and express the R-to-K mutant Sm proteins. |
| Transfected construct (*H. sapiens*) | Wild-type FLAG-CHTOP | VectorBuilder | VB210427-1238rhx | Lentiviral construct to transfect and express the wild-type CHTOP protein. |
| Transfected construct (*H. sapiens*) | R-to-A FLAG-CHTOP | VectorBuilder | VB210427-1241jrw | Lentiviral construct to transfect and express the R-to-A CHTOP protein. |
| Transfected construct (*H. sapiens*) | R-to-K FLAG-CHTOP | VectorBuilder | VB210427-1242gjh | Lentiviral construct to transfect and express the R-to-K CHTOP protein. |
| Transfected construct (*H. sapiens*) | dCas9-KRAB-MeCP2 | VectorBuilder | VB900120-5303pyt | Lentiviral construct to transfect and express dCas9-KRAB-MeCP2. |
| Transfected construct (*H. sapiens*) | gRNA parent vector | VectorBuilder | VB210119-1169qwd | Lentiviral construct to clone, transfect, and express PRMTkd gRNAs (see Materials and methods). |
| Antibody | Anti-human SNRPB (Rabbit polyclonal) | ProteinTech | Cat#: 16807-1-AP; RRID:AB_2878319 | WB: 1:2000 IF: 1:125 IP: 5 µg |
| Antibody | Anti-FLAG (Mouse monoclonal) | Sigma-Aldrich | Cat#: F1804; RRID:AB_262044 | WB: 1:10,000 IF: 1:50 IP: 5 µg |
| Antibody | Anti-Rabbit IgG (Goat polyclonal) | Thermo Fisher Scientific | Cat#: 35552; RRID:AB_844398 | IF: 1:1000 |
| Antibody | Anti-Mouse IgG (Goat polyclonal) | Thermo Fisher Scientific | Cat#: A28180 RRID:AB_2536164 | IF: 1:1000 |
| Antibody | Anti-Rme2s (Rabbit polyclonal) | CST | Cat#: 13222S RRID:AB_2714013 | WB: 1:2000 |
| Antibody | Anti-Rme2a (Rabbit polyclonal) | CST | Cat#: 13522S RRID:AB_2665370 | WB: 1:2000 |
| Antibody | Anti-Rme1 (Rabbit polyclonal) | CST | Cat#: 8015S RRID:AB_10891776 | WB: 1:2000 |
| Antibody | Anti-Mouse CHTOP (Rat monoclonal) | LSBio | Cat#: LS-B11259-50 | WB: 1:2000 IP: 5 µg |
| Antibody | Anti-human SNRPD3 (Rabbit polyclonal) | Abcam | Cat#: ab157118 | WB: 1:2000 |

*Continued on next page*

*Continued*

| Reagent type (species) or resource | Designation | Source or reference | Identifiers | Additional information |
|---|---|---|---|---|
| Antibody | Anti-human LaminB2 (Mouse monoclonal) | Thermo Fisher Scientific | Cat#: MA1-06104 RRID:AB_2136415 | WB: 1:2000 |
| Antibody | Anti-human U1-70k (Mouse monoclonal) | Santa Cruz Biotechnology | Cat#: sc-390988 | WB: 1:2000 |
| Antibody | Anti-human H3 (Rabbit polyclonal) | Abcam | Cat#: ab1791 RRID:AB_302613 | WB: 1:100,000 |
| Antibody | Anti-human GAPDH (Mouse monoclonal) | Abcam | Cat#: ab9484 RRID:AB_307274 | WB: 1:10,000 |
| Antibody | Anti-human PRMT1 (Rabbit polyclonal) | Millipore | Cat#: 07-404 RRID:AB_11212188 | WB: 1:2000 |
| Antibody | Anti-human PRMT5 (Rabbit polyclonal) | Millipore | Cat#: 07-405 RRID:AB_310589 | WB: 1:2000 |
| Antibody | Anti-Rat IgG HRP (Goat polyclonal) | Millipore | Cat#: AP183PMI | WB: 1:100,000 |
| Antibody | Anti-Rabbit IgG HRP (Goat polyclonal) | Cytiva | Cat#: NA934 | WB: 1:100,000 |
| Antibody | Anti-Mouse IgG HRP (Goat polyclonal) | Cytiva | Cat#: NA931 | WB: 1:100,000 |
| Chemical compound, drug | DMSO | Acros Organics | Cat#: D/4125/PB08 | |
| Chemical compound, drug | GSK591 | Cayman | Cat#: 18354 | |
| Chemical compound, drug | MS023 | Cayman | Cat#: 18361 | |
| Chemical compound, drug | Actinomycin D | Sigma-Aldrich | Cat#: A1410 | |
| Software, algorithm | STAR | *Dobin et al., 2013* | RRID:SCR_004463 | Version 2.4.2 |
| Software, algorithm | rMATS | *Shen et al., 2014* | RRID:SCR_013049 | Version 4.1.0 |
| Software, algorithm | Kallisto | *Bray et al., 2016* | RRID:SCR_016582 | Version 0.46.0 |
| Software, algorithm | Proteome Discoverer software | Thermo Fisher Scientific | | Version 2.4 |
| Software, algorithm | Prism | Graphpad | RRID:SCR_002798 | |
| Software, algorithm | R | R Project | RRID:SCR_001905 | Version 4.0.2 |
| Other | DAPI Prolong gold | Invitrogen | Cat#: P36941 | |

## Cell culture

A549 and HEK293T cells were cultured in DMEM (Corning) supplemented with 10% FBS (Hyclone), 100 µg/ml streptomycin, and 100 I.U./ml penicillin (Corning) and maintained at 37°C with humidity and 5% $CO_2$. For this study, fresh cells were purchased from ATCC and tested routinely for *Mycoplasma* (PCR, FWD primer: ACTCCTACGGGAGGCAGCAGT, REV primer: TGCACCATCTGTCACT CTGTTAACCTC) (*Dussurget and Roulland-Dussoix, 1994*).

## RT-qPCR

RNA purification was performed using RNeasy Plus (QIAGEN) or TRIzol (Thermo Fisher Scientific). Isolated RNA was reverse transcribed with Moloney Murine Leukemia Virus (MMLV) reverse transcriptase (Invitrogen) and oligo(dT) primers. LightCycler 480 Sybr Green I (Roche) master mix was used to quantitate cDNA with a LightCycler 480 (Roche). An initial 95°C for 5 min was followed by 45 cycles

of amplification using the following settings: 95°C for 15 s, 60°C for 1 min. Primer sequences can be found in *Supplementary file 2*.

## Poly(A)-RNA sequencing

RNA was extracted using RNeasy Mini Kit (QIAGEN) following the manufacturer's protocol. RNA quantitation and quality control were accomplished using the Bioanalyzer 2100 (Agilent Technologies). Stranded RNA seq libraries were constructed by Novogene Genetics US. The barcoded libraries were sequenced by Novogene on an Illumina platform using 150 nt paired-end libraries generating ~30–40 million reads per replicate. Reads were trimmed and aligned to the human genome (hg19) with Spliced Transcripts Alignment to a Reference (STAR) (*Dobin et al., 2013*). Alternative splicing events were determined using rMATS (version 4.0.2) (*Shen et al., 2014*). Expression was determined using Kallisto (*Bray et al., 2016*). IGV (Broad Institute) was used as the genome browser. Graphs pertaining to RNA seq were created using Gviz (*Hahne and Ivanek, 2016*) in R (4.0.2) and assembled in Adobe Illustrator 2020.

## Splicing kinetics and transcript elongation rates by sequencing

SKaTER seq was performed as described (*Casill et al., 2021*). Briefly, A549 cells were grown with 0.01% DMSO, 1 µM GSK591 (Cayman), or 1 µM MS023 (Cayman) for 2 days followed by addition of 100 µM DRB (Cayman). DRB-containing media was removed, and the cells were incubated at 37°C until the indicated time point. The cells were washed once with 4°C phosphate-buffered saline (PBS), and lysed by addition of 1 ml CL buffer (25 mM Tris pH 7.9 at 4°C, 150 mM NaCl, 0.1 mM EDTA, 0.1% Triton X-100, and 1 mM DTT) supplemented with protease inhibitor (Thermo Fisher Scientific) and *Drosophila melanogaster* S2 cell spike-in. Next, lysate was centrifuged at 845×$g$ for 5 min at 4°C. The pellet resuspended in 1 ml CL buffer without S2 spike-in and incubated on ice for 5 min. Repeat centrifugation was performed. The supernatant was removed, and cells resuspended in 100 µl GR buffer (20 mM Tris pH 7.9, 75 mM NaCl, 0.5 mM EDTA, 50% glycerol, and 0.85 mM DTT) followed by addition of 1.1 ml NL buffer (20 mM HEPES pH 7.6, 300 mM NaCl, 7.5 mM MgCl$_2$, 1% NP-40, 1 mM DTT, and 1 M Urea). Following a 15-min incubation, the lysate was spun at 16,000×$g$ for 10 min and the resulting chromatin pellet was resuspended and stored in TRIzol (Thermo Fisher Scientific) at –80°C. RNA isolation was followed by poly(A) depletion using the NEBNext Poly(A) RNA magnetic isolation module (NEB). RNA quantitation and quality control were accomplished using the Bioanalyzer 2100 (Agilent Technologies). Stranded RNA-seq libraries were prepared using the KAPA RNA HyperPrep Kit with RiboErase (HMR) and KAPA Unique Dual-Indexed Adapters (Roche) according to instructions provided by the manufacturer. The barcoded paired-end libraries were sequenced by Novogene using a NovaSeq S4, generating ~70 million reads per replicate.

## Actinomycin D post-transcriptional processing

A549 cells were grown in the presence of 0.01% DMSO, 1 µM GSK591 (Cayman), or 1 µM MS023 (Cayman) for 2 days. Following a 2-day incubation, the media were removed and replaced with media containing 5 µg/ml actinomycin D (Sigma-Aldrich) with 0.01% DMSO, 1 µM GSK591, or 1 µM MS023 for 60 min. RNA was isolated using RNeasy Mini Kit (QIAGEN) and poly(A)-RNA sequencing was performed as above.

## Chromatin-associated poly(A)-RNA enrichment and LC-MS/MS

Poly(A)-RNA isolation was performed with modifications to a previously described protocol (*Iadevaia et al., 2018*). A549 cells were grown in the presence of 0.01% DMSO, 1 µM GSK591 (Cayman), or 1 µM MS023 (Cayman) for 2 days. Cells were washed with 4°C PBS and irradiated on ice with 100 mJ cm$^{-2}$ in a UV Stratalinker 1800. Cells were centrifuged at 500×$g$ for 10 min at 4°C. Chromatin was isolated with nuclear lysis buffer (NLB; 10 mM Tris-HCl pH 7.5 at 4°C, 0.1% NP-40, 400 mM KCl, and 1 mM DTT) supplemented with 40 U/ml RNaseOUT (Thermo Fisher Scientific), protease inhibitor (Thermo Fisher Scientific), and phosphatase inhibitor (Thermo Fisher Scientific). The chromatin pellet was resuspended in NLB and sonicated for 5 s at 20% amplitude with a probe-tip sonicator using a 1/8 inch tip. The sonicate was centrifuged at 10,000×$g$ for 10 min and the soluble material transferred to a low-adhesion RNase-free microcentrifuge tube. An aliquot from each sample was saved to serve as the unenriched control. The samples were split into two separate tubes, one of which received

10 µg of competitor 25-nt poly(A) RNA. Magnetic oligo(dT) beads (NEB) were equilibrated in NLB and added to the enrichments. The samples were vortexed at room temperature for 10 min. The beads were then captured on a magnetic column, and the supernatant transferred to fresh tube for additional rounds of depletion. The beads were washed once with buffer A (10 mM Tris pH 7.5, 600 mM KCl, 1 mM EDTA, and 0.1% Triton X-100), followed by buffer B (10 mM Tris pH 7.5, 600 mM KCl, and 1 mM EDTA) and finally buffer C (10 mM Tris pH 7.5, 200 mM KCl, and 1 mM EDTA). The RNA was eluted by incubating the beads in 10 µl of 10 mM Tris pH 7.5 at 80°C for 2 min, capturing the magnetic beads with a magnetic column, and quickly transferring the supernatant to a new tube. The beads were then used for two additional rounds of poly(A)-RNA capture.

The (un)enriched proteome were treated with Benzonase (Sigma-Aldrich) and then digested with trypsin (Promega) prior to performing LC-MS/MS. Proteins were resuspended in 5% SDS and 5 mM DTT and left incubating for 1 hr at room temperature. Samples were then alkylated with 20 mM iodoacetamide in the dark for 30 min. Afterward, phosphoric acid was added to the sample at a final concentration of 1.2%. Samples were diluted in six volumes of binding buffer (90% methanol and 10 mM ammonium bicarbonate, pH 8.0). After gentle mixing, the protein solution was loaded to an S-trap filter (Protifi) 96-well plate and spun at 500×$g$ for 30 s. The sample was washed twice with binding buffer. Finally, 100 ng of sequencing grade trypsin (Promega)—diluted in 50 mM ammonium bicarbonate—was added into the S-trap filter and samples were digested at 37°C for 18 hr. Peptides were eluted in three steps: (i) 40 µl of 50 mM ammonium bicarbonate, (ii) 40 µl of 0.1% TFA, and (iii) 40 µl of 60% acetonitrile and 0.1% TFA. The peptide solution was pooled, spun at 1000×$g$ for 30 s, and dried in a vacuum centrifuge. Prior to mass spectrometry analysis, samples were desalted using a 96-well plate filter (Orochem) packed with 1 mg of Oasis HLB C-18 resin (Waters). Briefly, the samples were resuspended in 100 µl of 0.1% TFA and loaded onto the HLB resin, which was previously equilibrated using 100 µl of the same buffer. After washing with 100 µl of 0.1% TFA, the samples were eluted with a buffer containing 70 µl of 60% acetonitrile and 0.1% TFA and then dried in a vacuum centrifuge. For LC-MS/MS acquisition, samples were resuspended in 10 µl of 0.1% TFA and loaded onto a Dionex RSLC Ultimate 300 (Thermo Fisher Scientific), coupled online with an Orbitrap Fusion Lumos (Thermo Fisher Scientific). Chromatographic separation was performed with a two-column system, consisting of a C-18 trap cartridge (300 µm ID, 5 mm length) and a picofrit analytical column (75 µm ID, 25 cm length) packed in-house with reversed-phase Repro-Sil Pur C18-AQ 3 µm resin. Peptides were separated using a 60-min gradient from 4% to 30% buffer B (buffer A: 0.1% formic acid, buffer B: 80% acetonitrile + 0.1% formic acid) at a flow rate of 300 nL/min. The mass spectrometer was set to acquire spectra in a data-dependent acquisition (DDA) mode. The full MS scan was set to 300–1200 m/z in the orbitrap with a resolution of 120,000 (at 200 m/z) and an AGC target of 5×$10^5$. MS/MS was performed in the ion trap using the top speed mode (2 s), an AGC target of $10^4$ and an HCD collision energy of 35. Raw files were searched using Proteome Discoverer software (v2.4, Thermo Fisher Scientific) using SEQUEST search engine with the SwissProt human database. The search for total proteome included variable modification of N-terminal acetylation and fixed modification of carbamidomethyl cysteine. Trypsin was specified as the digestive enzyme with up to two missed cleavages allowed. Mass tolerance was set to 10 pm for precursor ions and 0.2 Da for product ions. Peptide and protein false discovery rate was set to 1%.

Data were transformed, normalized and statistics were applied as described previously (*Aguilan et al., 2020*). Briefly, the data were $log_2$ transformed and normalized by the average of the data distribution. Statistical analysis was performed by using a two-tail heteroscedastic t-test.

## CHTOP and SNRPB immunoprecipitation

A549 cells were grown with 0.01% DMSO, 1 µM GSK591 (Cayman), or 1 µM MS023 (Cayman) for 2 days. Cells were harvested using trypsin (Corning) and washed once with 4°C PBS supplemented with PRMT inhibitors. Cells were then resuspended in RIPA buffer (1% NP-40, 150 mM NaCl, 50 mM Tris-HCl pH 8 at 4°C, 0.25% sodium deoxycholate, 0.1% SDS, and 1 mM EDTA) supplemented with 40 U/mL RNaseOUT (Thermo Fisher Scientific) and protease inhibitor (Thermo Fisher Scientific). Lysates were incubated on ice for 10 min followed by sonication for 5 s at 20% amplitude with a probe-tip sonicator using a 1/8 inch tip. Lysates were then spun at 10,000×$g$ for 10 min at 4°C. Supernatants were transferred to new low-adhesion RNase-free microcentrifuge tubes and normalized to the same protein concentration using bicinchoninic acid (Pierce). Primary antibody targeting CHTOP (LSBio,

LS-C193506; 5 µg) or SNRPB (ProteinTech, 16807-1-AP; 5 µg) was added followed by incubation over-night at 4°C with gentle rotation. The next morning, Protein G agarose (Millipore-Sigma) was equilibrated in lysis buffer and added to the lysates at 4°C with gentle rotation. The beads were washed three times with lysis buffer followed by resuspension in 1× Laemmli buffer for western blotting with CHTOP (as above), SNRPB (as above), Rme2s (CST, 13222), and Rme2a (CST, 13522) antibodies.

## CRISPRi and expression of methylarginine mutants

The dCas9-KRAB-MeCP2 expression vector (VB900120-5303pyt) was purchased from VectorBuilder (Santa Clara, CA). The PRMT-targeting gRNA parent vector (VB210119-1169qwd) was purchased from VectorBuilder and custom gRNA sequences (*Supplementary file 2*) derived from CRISPick (*Doench et al., 2016*) were cloned as described previously (*Ran et al., 2013*) with the exception that BfuA1 (NEB) was used to digest the parent vector. Sm WT (VB210103-1026dg), R-to-A (VB210103-1027ttz), and R-to-K (VB210317-1185yxr) as well as CHTOP WT (VB210427-1238rhx), R-to-A (VB210427-1241jrw), and R-to-K (VB210427-1242gjh) expression vectors were cloned by VectorBuilder. Lentiviral particles containing expression vectors were produced in HEK293T cells using calcium phosphate transfection. Transduction of A549 cells was accomplished by combining lentivirus with polybrene containing media (4 µg/ml) and centrifuging at 30°C for 90 min at 500×$g$ followed by incubation for 24 hr at 37°C, 5% CO$_2$ with humidity. Following the 24-hr incubation, lentiviral containing media was removed and complete DMEM containing appropriate selection antibiotic (Cayman) was added. A549 cells expressing dCas9-KRAB-MeCP2 were selected using 2 µg/ml puromycin and then maintained in culture with 1 µg/ml puromycin. For PRMT knockdown and Sm expression experiments, A549 cells were selected with 10 µg/ml blasticidin for 72 hr. For CHTOP expression experiments, A549 cells were selected for 96 hr with 1 mg/ml G418. Primers used in RT-qPCR experiments can be found in *Supplementary file 2*. Lysis for western blotting was accomplished as above and the antibodies used included Rme2s (CST, 13222), Rme2a (CST, 13522S), GAPDH (Abcam, ab9484), PRMT1 (Millipore, 07-404), PRMT5 (Millipore, 07-405), and FLAG (Sigma-Aldrich, F1804).

## Cell fractionation

A549 cells were grown with 0.01% DMSO, 1 µM GSK591 (Cayman), or 1 µM MS023 (Cayman) for 2 days. Cells were harvested using trypsin (Corning) and washed once with 4°C PBS. Cells were resuspended in Hypotonic Lysis Buffer (10 mM Tris-Cl pH 8 at 4°C, 0.1% NP-40, 1 mM KCl, 1.5 mM MgCl$_2$, 1 mM DTT supplemented with protease inhibitor and 40 U/ml RNaseOUT) and rotated for 30 min at 4°C. Cells were then centrifuged at 10,000×$g$ for 10 min at 4°C and the supernatant was kept as the cytosolic fraction. The nuclear pellet was washed once with hypotonic buffer using a wide-orifice tip and then resuspended in NLB (10 mM Tris-HCl pH 8 at 4°C, 0.1% NP-40, 400 mM KCl, 1 mM DTT supplemented with protease inhibitor, and 40 U/mL RNaseOUT) followed by rotation for 30 min at 4°C. The sample was then centrifuged at 10,000×$g$ for 10 min and the supernatant was kept as the nucleoplasm, while the remaining material was washed once with NLB and kept as the chromatin fraction. RNA from each fraction was isolated using TRIzol (Thermo Fisher Scientific) and RT-qPCR was performed as above. For western analysis, 10% of each fraction was set aside and diluted to 1× with Laemmli buffer. To isolate the chromaitn fraction for western analysis, after resuspension of nuclei in NLB, 10% was aliquoted into a low-adhesion microcentrifuge tube. Following centrifugation, the supernatant was saved as the nucleoplasm. The chromatin fraction was subsequently washed once with NLB and then resuspended again in NLB equivalent to the original volume and sheared with sonication. This was followed by addition of Laemmli buffer to 1× and western blotting with H3 (Abcam, ab1791), GAPDH (Abcam, ab9484), U1-70k (Santa Cruz Biotechnology, sc-390988), LaminB2 (Thermo Fisher Scientific, MA1-06104), CHTOP (LSBio, LS-B11259-50), Rme2a (CST, 13522), SNRPB (Protein-Tech, 16807-1-AP), SNRPD3 (Abcam, ab157118), and Rme2s (CST, 13222) antibodies.

## FLAG co-immunoprecipitation

A549 cells were transduced with Sm expression constructs as above. Cells were harvested using trypsin (Corning), washed once with 4°C PBS, and then resuspended in NP-40 Lysis Buffer (0.5% NP-40, 50 mM Tris pH 8 at 4°C, 150 mM NaCl, 1 mM EDTA) supplemented with 40 U/ml RNaseOUT (Thermo Fisher Scientific) and protease inhibitor (Thermo Fisher Scientific). Lysates were incubated on ice for 10 min followed by sonication for 5 s at 20% amplitude with a probe-tip sonicator using a 1/8

inch tip. Lysates were then spun at 10,000×g for 10 min at 4°C. Supernatants were transferred to new low-adhesion RNase-free microcentrifuge tubes and pre-cleared for 30 min with Protein G agarose (Millipore-Sigma) equilibrated in lysis buffer. The pre-cleared supernatants were again transferred to new low-adhesion RNase-free microcentrifuge tubes and normalized to the same protein concentration using Bradford assay (Bio-Rad). Primary antibody FLAG (Sigma-Aldrich, F1804; 5 µg), SNRPB (ProteinTech, 16807-1-AP; 5 µg), or control IgG (Abcam, ab46540; 5 µg) was added and incubated overnight at 4°C with gentle rotation. The next morning, Protein G agarose was equilibrated in lysis buffer and added to the lysates at 4°C with gentle rotation for 2 hr. The beads were washed three times with lysis buffer containing 300 mM NaCl followed by resuspension in either 1× Laemmli buffer for western blotting or TRIzol (Thermo Fisher Scientific) for northern blotting.

## Northern blotting

RNA was isolated using TRIzol (Thermo Fisher Scientific) and coprecipitated with GlycoBlue (Thermo Fisher Scientific). The RNA pellet was resuspended in sample buffer (6.8 M Urea in TBE with 10% glycerol and 0.25% Bromophenol Blue/Xylene Cyanide), heated at 90°C for 3 min, followed by loading onto an 8% Urea Gel (National Diagnostics) that was pre-run for 45 min at 45 W. The gel was run for 1 hr at 45 W in 1× TBE (100 mM Tris, 100 mM Boric acid, and 0.2 mM EDTA) followed by transfer to nitrocellulose in 0.5× TBE at 30 mA for 4 hr at 4°C. Following transfer, RNA was crosslinked to the membrane at 120,000 µJ/cm$^2$ (UV Stratalinker 1800). 5′ end labeling of snRNA probes (*Supplementary file 2*) was performed using ATP [$^{32}$P] (PerkinElmer) with T4 PNK reaction (NEB). Unincorporated ATP [$^{32}$P] was removed using a Microspin G-25 column (Cytiva). Post-transfer hybridization was performed at 37°C in a hybridization oven overnight with gentle agitation in hybridization buffer containing 100 mM NaHPO$_4$ pH 7.2, 750 mM NaCl, 1× Denhardt's Solution (0.02% BSA, 0.02% Ficoll 400, and 0.02% Polyvinylpyrrolidone), 1% Herring sperm DNA, and 7% SDS. The next morning, the hybridization solution was carefully discarded according to institutional protocol and the membrane was washed twice with wash buffer (40 mM NaHPO$_4$ pH 7.2, 2% SDS, 1 mM EDTA). The wash buffer was also discarded according to institutional guidelines. The membrane was left to expose on a Phosphoimager screen (Cytiva) and imaged at 633 nm using a Typhoon 9400 Variable Mode Imager.

## Indirect immunofluorescence

Cells were seeded on coverslips (Corning) and either transduced with Sm proteins (as above) or allowed to grow in the presence of 0.01% DMSO, 1 µM GSK591 (Cayman), or 1 µM MS023 (Cayman) for 2 days. Cells were then washed with 37°C PBS (Hyclone) and fixed with 4% paraformaldehyde at 20–25°C for 10 min followed by washing with 4°C PBS. Residual aldehyde was quenched with 0.1 M glycine in 20–25°C PBS for 15 min. Permeabilization was performed using 0.1% Triton X-100 at 20–25°C with gentle rotation for 30 min. Cover slips were then washed with PBS and blocked for one hour using 0.1% Fish Skin Gelatin in PBS. Primary antibody for FLAG (Sigma-Aldrich, F1804; 1:50) or SNRPB (ProteinTech 16807-1-AP; 1:125) was added to the cover slip and incubated overnight at 4°C in blocking buffer. Coverslips were washed with PBS and incubated with secondary antibody (Goat anti-Rabbit Dylight 488, Thermo Fisher Scientific 35552 or Goat anti-Mouse Alexa Fluor 555, Thermo Fisher Scientific A28180; 1:1000) with gentle rotation while protected from light for 1 hr at 20–25°C followed by PBS wash and mounting with DAPI prolong gold anti-fade (Thermo Fisher Scientific). Imaging was performed with an Olympus IX-70 inverted microscope with a 60× objective.

## Statistical analysis

All western blots were performed independently at least twice. RT-qPCR was performed at least three times with independent biological replicates. Statistical analyses were performed using either Prism software (version 8.3.1, GraphPad) or R (version 4.0.2). To compare general distributions, the Kolmogorov-Smirnov test was used. To compare median distribution changes, the Wilcoxon rank-sum test was used. To account for differences in sample size between global and RI distributions, random sampling from the global population equivalent to the number of RI within the tested condition was performed. This process was repeated 1000 times after which the median p-value was reported. To compare means where only two groups exist, independent t-test was performed. To compare means where two categorical variables exist, two-way ANOVA with Tukey's multiple comparisons test was performed. GeneOverlap with Fisher's exact test was used to determine ORs of RI overlap between

different RNA-seq data sets (*Shen, 2020*). All code used to generate data in this manuscript can be found here: https://github.com/Shechterlab/PRMTsRegulatePostTranscriptionalDI [copy archived at https://doi.org/10.5281/zenodo.5851791 (*Maron, 2022*)].

## Acknowledgements

The authors thank Dr. Martin Krzywinski (http://mkweb.bcgsc.ca/) for help with data visualization.

## Additional information

### Funding

| Funder | Grant reference number | Author |
| --- | --- | --- |
| National Institute of General Medical Sciences | R01GM108646 | David Shechter |
| National Institute of General Medical Sciences | R01GM57829 | Charles C Query |
| National Institute of General Medical Sciences | R01GM134379 | Matthew J Gamble |
| American Lung Association | LCD-564723 | David Shechter |
| Irma T. Hirschl Trust | | Matthew J Gamble David Shechter |
| National Institutes of Health | S10OD030286 | Simone Sidoli |
| National Cancer Institute | P30CA013330 | Simone Sidoli |

The funders had no role in study design, data collection and interpretation, or the decision to submit the work for publication.

### Author contributions

Maxim I Maron, Conceptualization, Data curation, Formal analysis, Investigation, Methodology, Validation, Visualization, Writing – original draft, Writing – review and editing; Alyssa D Casill, Formal analysis, Software; Varun Gupta, Formal analysis; Jacob S Roth, Investigation; Simone Sidoli, Formal analysis, Methodology, Writing – review and editing; Charles C Query, Matthew J Gamble, Experimental design and data interpretation, Experimental design and data interpretation, Funding acquisition, Writing – review and editing; David Shechter, Conceptualization, Funding acquisition, Project administration, Supervision, Writing – original draft, Writing – review and editing

### Author ORCIDs

Maxim I Maron http://orcid.org/0000-0001-7809-5888
Varun Gupta http://orcid.org/0000-0001-5330-0486
Jacob S Roth http://orcid.org/0000-0003-0521-8677
Simone Sidoli http://orcid.org/0000-0001-9073-6641
Charles C Query http://orcid.org/0000-0002-7692-2496
Matthew J Gamble http://orcid.org/0000-0002-8107-0523
David Shechter http://orcid.org/0000-0001-9388-6004

### Decision letter and Author response

Decision letter https://doi.org/10.7554/eLife.72867.sa1
Author response https://doi.org/10.7554/eLife.72867.sa2

## Additional files

### Supplementary files

• Supplementary file 1. Chromatin-Associated Poly(A) RNA Enrichment. Enrichment and relative abundance of proteins in chromatin or chromatin-associated poly(A) RNA following treatment with

DMSO, GSK591, or MS023 for two days.

- Supplementary file 2. Oligonucleotide sequences used in this study.
- Supplementary file 3. Publicly available data used in this study.
- Transparent reporting form

## Data availability

Raw data for RNA seq and SKaTER seq is deposited under GEO (GSE163421) Raw data for chromatin-associated poly(A) LC-MS/MS is deposited under Chorus (1729). All code used to generate data in this manuscript can be found here: https://github.com/Shechterlab/PRMTsRegulatePostTranscriptionalDI (copy archived at https://doi.org/10.5281/zenodo.5851791).

The following dataset was generated:

| Author(s) | Year | Dataset title | Dataset URL | Database and Identifier |
|---|---|---|---|---|
| Maron MI, Casill AD, Gupta V, Sidoli S, Query CC, Gamble MJ, Shechter D | 2020 | Type I and II PRMTs Inversely Regulate Post-Transcriptional Intron Detention through Sm and CHTOP methylation | https://www.ncbi.nlm.nih.gov/geo/query/acc.cgi?acc=GSE163421 | NCBI Gene Expression Omnibus, GSE163421 |

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
