## [Decision Letter]

**Decision letter after peer review:**

Thank you for submitting your article "Type I PRMTs and PRMT5 Inversely Regulate Post-Transcriptional Intron Detention" for consideration by *eLife*. Your article has been reviewed by 3 peer reviewers, one of whom is a member of our Board of Reviewing Editors, and the evaluation has been overseen by James Manley as the Senior Editor. The reviewers have opted to remain anonymous.

The reviewers have discussed their reviews with one another, and the Reviewing Editor has drafted this letter to help you prepare a revised submission.

Overall, the reviewers were excited about the potential insight from this study, but agreed on several additional experiments that are necessary to support the conclusions.

Essential revisions:

1) First, the effectiveness and specificity of the chemical PRMT inhibitors needs to be confirmed by western blot or mass spectrometry of known specific targets

2) Secondly, it was agreed that the use of transcriptional inhibition to suggest post-transcriptional regulation is problematic in that transcriptional inhibition induces a cellular stress response. In order to make the important conclusion about post-transcriptional regulation of splicing the authors should sequence (or assess the splicing of a representative set of candidate genes) RNA from chromatin, soluble nucleoplasmic and cytoplasmic fractions. Such an analysis will also provide information on the regulation of splicing vs. export.

3) Regarding the conclusions about the functional impact of methylation of Sm proteins the authors should determine if the mutant proteins are incorporated into snRNPs and/or if there is a general impact on snRNP levels (reviewer 1 point 3) and also investigate the nucleo/cytoplasmic distribution of SmB to determine if methylation state regulates localization (reviewer 2).

*Reviewer #1 (Recommendations for the authors):*

1) Show deltaPSI in Figure 1e not log(PRMT/DMSO) to be consistent and to clearly indicate the differences seen.

2) I don't understand the rationale of the statement "a slower splicing rate should increase RI" (line 188). Isn't it possible that a slower rate increases fidelity and thus increases intron removal at steady state, whereas rapid splicing might "miss" some introns? Maybe I'm wrong, but I'm not sure it is fair to conclude that splicing rate leads to the conclusion that regulation by type I PRMTs is post-transcriptional.

3) In the Sm mutational experiments, are the R-A mutants incorporated into snRNPs? That is, is the impact due indirectly to a decreased number of snRNPs or a direct effect on the activity of the snRNPs?

*Reviewer #2 (Recommendations for the authors):*

Figure 1: The inverse splicing of many RIs in response to the different inhibitors is evident in 1c, but these numbers add up to less than half of the introns affected by MS023 shown in f and g. If you measure these exons as a population (rather than comparing them one-for one) in the GSK591 treatment vs. the MS023 treatment, does the distribution move significantly in the opposite direction? The question is whether this subset is responding in the same inverse manner but is below significance threshold, or whether these may represent some indirectly affected second group.

Figure 2: This figure is difficult to follow, although along with Figure 3 makes the important observation that these effects are most likely post-transcriptional, given that the expected result for co-transcriptional effects would be the opposite of what is seen with MS023 treatment. Specifically, 2 c and d and 3 e are the critical pieces of data. Rearranging the order of the figure panels and potentially moving the more trivial results into supplemental might help clarify the presentation.

The measurement of splicing rates and the here is straightforward and directly determined by the experimental data. I'm less convinced that the PSI simulation in panel b is not a circular process, and more direct experimental data is required to make the argument that alternative splicing ratios are so universally and profoundly affected by elongation rates.

The fact that the data in panels g and h are contradictory to the elongation rate model of splicing (since splicing rates should be determined by the overall time to transcribe, which is significantly faster here, even though measured elongation and splicing rates are lower, in both treatments) supports the conclusion that the differential effects of PRMTi occurs post-transcriptionally. The current presentation of the data in the text, especially the concluding sentence (line 211 pg. 11) "Taken together, these results support the conclusion that RI in GSK591- and MS023-treated cells share unique characteristics that increase their probability of post-transcriptional processing" should more explicitly state that the unique characteristics referred to are that they don't behave according to the model-as written, the reader spends a lot of time trying to figure out how these seemingly contradictory data support the conclusion.

Figure 3: This is a difficult experiment, but really pins down the post-transcriptional effect on intron abundance.

Figure 4: It would be potentially very informative to see the soluble (nucleoplasmic) and cytoplasmic fractions of these blots-is there a pool of methylated SNRPB in steady-state that moves to the chromatin fraction upon demethylation (in other words, is the methylation state likely controlling the localization of the protein). The same fractionation on the arginine substitution mutants would further address the question.

Figure 7: The only part of the model that is not directly supported by the data here is the inhibition of RNA export and the promotion of post-transcriptional splicing via PRMT1 inhibition. If the observation were of spliced mRNA remaining nuclear in the presence of PRMT1 inhibition, that pathway could be inferred. However, in the absence of export, even fully-spliced mRNA might be turned over quickly. Alternatively, PRMT1 inhibition could result in a failure to protect DI-containing transcripts, which are then turned over in the nucleus. Dissecting this part of the pathway will likely require further experiments involving nuclear exosome inactivation.

*Reviewer #3 (Recommendations for the authors):*

I have several concerns that I feel need to be addressed before settling on these interpretations, including details of particular protein activities invoked, e.g. by CHTOP, where the experimental procedures do not necessarily match the question posed and/or the conclusions drawn.

1. The manuscript is really poorly written and hard to understand. The language is unnecessarily complicated and involves a lot of double negatives that seem unnecessary. For example, lines 15-18 in the abstract are literally impossible to understand. On line 22 "Conversely" is not understandable, because the results do recapitulate the splicing changes seen (as do the results in the previous sentence); "similarly" or "in addition" might be appropriate. This type of confusing language choice is pervasive.

2. The use of the term PRMTi obscures clarity about the specific experiments, which are done mostly with chemical inhibitors. In fact, the sentence "We also examine which factors are responsible for PRMTi-mediated splicing consequences" (lines 75-76) suggests PRMTi is a normal physiological pathway, which it is not. I strongly suggest the authors remove PRMTi as a term used in the paper. A mostly supplemental section uses a CRISPR interference strategy taken for validation, which was less robust that chemical inhibitor treatments. The authors must investigate the effects of chemical inhibition and CRISPR interference methods on dimethylarginine levels in order to correctly draw their conclusions! Other labs have reported only modest effects on sDMA and aDMA after even longer incubations.

Related to the above issue of language, the statement on lines 301-2 "…. this suggested that SNRPB may also be involved in the regulation of GSK591-mediated RI." This is so confusing for the reader. Its sounds like SNRPB is an alternative splicing factor involved in regulating something that is not physiological. What does this sentence mean?

3. GSK591 inhibits PRMTs 5 and 9. It is stated on line 90 that "PRMT5 is the primary Type II methyltransferase (Yang et al., 2015)", and Type II is dropped from discussion for the remainder of the manuscript. For example, the figure title for Figure 1 is "Type I PRMTs and PRMT5 inversely regulate intron retention". Note that Yang et al. show that PRMT9 methylates SAP145 (SF3B2) and that knockdown affects splicing; SAP145 is a protein component of the U2 snRNP that is required for A complex formation. Therefore, ignoring PRMT9 is not justifiable. The authors should refer to Type II PRMTs throughout the text and in the title of the paper. As it stands, the title is not justified by the data.

4. Post-transcriptional splicing is investigated by inhibiting transcription in Figure 3. It is becoming increasingly clear that wholesale chemical inhibition of transcription or splicing can lead to a generalized stress response that may not reflect the "normal" unstressed situation. Would it not be possible to conduct their experiment in nucleoplasm versus chromatin, as was done by others and analyzed in Figure 6? This is especially frustrating, given the authors do do the fractionation in the context of figure 4.

5. In figure 4, the authors identify a post-translational mRNA interactome by isolating polyA+ RNA from chromatin. Overall, this is a really great undertaking with potentially high impact. However, few of the dots are connected for us in the context of this experiment. The rationale is that previous studies suggest retention of transcripts on chromatin. The interpretation of this figure is that PRMT inhibitors shift proteins aberrantly in and out of the chromatin-proximal polyA+ pool. To make this interpretation, it would be necessary to analyze all of the polyA+ interacting proteins in nucleoplasm and compare them with the pool of interactors in the chromatin fraction. Otherwise, other less direct interpretations could come into play. It is unclear how specific the polyA+ RNA isolated in this experiment is; how can we distinguish between background binding and polyA+ RNA specifically retained on chromatin due to continued processing? It would be great if the authors undertook some orthogonal experiments, e.g. through imaging, to characterize the localization patterns of snRNPs and CHTOP relative to one another and to chromatin/nascent transcripts in the presence and absence of the inhibitors.

---

## [Author Response]

Essential revisions:1) First, the effectiveness and specificity of the chemical PRMT inhibitors needs to be confirmed by western blot or mass spectrometry of known specific targets

We agree that validating the effectiveness and specificity of the chemical PRMT inhibitors is crucial. We recently published an extensive mass spectrometry analysis of PRMT inhibition using the same inhibitors and the same A549 cells used in this study (Maron et al., iScience 2021, https://doi.org/10.1016/j.isci.2021.102971). We now clearly cite this work in the current paper to validate the use of the drugs, including on our main candidate proteins SNRPB/SmB and CHTOP. We also added new experiments comparing drug treatment to the PRMT knockdowns (Figure 1 —figure supplement 1). Finally, these PRMT inhibitors have been widely used and validated in many papers, which we now reference in the main text. Overall, we have strong support for the use of these inhibitors in this study.

2) Secondly, it was agreed that the use of transcriptional inhibition to suggest post-transcriptional regulation is problematic in that transcriptional inhibition induces a cellular stress response. In order to make the important conclusion about post-transcriptional regulation of splicing the authors should sequence (or assess the splicing of a representative set of candidate genes) RNA from chromatin, soluble nucleoplasmic and cytoplasmic fractions. Such an analysis will also provide information on the regulation of splicing vs. export.

We addressed this critique in the revised Figure 6. In these new experiments, we performed the requested experiment by testing four introns (including a new representative RI in HNRNPH1) after fractionation into cytoplasmic, soluble nuclear, and chromatin fractions, with and without prior treatment with actinomycin D. As the corresponding gain or loss of intron retention is only found in the nuclear/chromatin fractions, the new data support our conclusion. Due to cost and time constraints (3 drugs in 3 fractions, with and without actinomycin D in triplicate is 54 new samples, making it currently cost and time prohibitive), we did not sequence these experiments. Our probed introns are representative and consistent throughout the manuscript.

Regarding the cellular stress response, there is no simple way to probe nascent and post-transcriptional consequences without the use of a transcriptional inhibitor like actinomycin D or a labeled ribonucleotide (which could also result in a stress response). It is also unclear from the literature how to mechanistically or conceptually separate stress response from specifically regulated consequences. As this is a broad subject outside the scope of the current manuscript, future studies are required to disentangle a general stress response from PRMT-mediated consequences.

3) Regarding the conclusions about the functional impact of methylation of Sm proteins the authors should determine if the mutant proteins are incorporated into snRNPs and/or if there is a general impact on snRNP levels (reviewer 1 point 3) and also investigate the nucleo/cytoplasmic distribution of SmB to determine if methylation state regulates localization (reviewer 2).

We thank the reviewers for prompting us to strengthen the data in these studies. To test the functional role of methylarginine in Sm protein / snRNP assembly we added the following experiments to the manuscript:

1) We expressed Sm WT, RA, RK mutants and FLAG IP’d followed by western blotting for U1-70k, SMN, FLAG, and Rme2s;

2) In the FLAG IP’d material, we also northern blotted for U1, U2, U4, U5, and U6 snRNA and compared this to SmB IP’d snRNA;

3) We treated cells with the PRMT inhibitors and IP’d SmB followed by northern blotting for U1, U2, U4, U5, and U6 snRNA;

4) We expressed Sm WT, RA, RK mutants and performed indirect immunofluorescence for FLAG and SmB;

5) We performed indirect immunofluorescence for SmB in PRMT inhibited cells;

6) We western blotted for SmB and SmD3 across cytoplasmic, nucleoplasmic, and chromatin fractions following PRMT inhibition.

Together, these experiments demonstrate that despite having no detectable Rme2s on the mutant Sm proteins, as well as reduced Rme2s with the Type II inhibitor, the Sm-mutant and PRMT-inhibited cells all have appropriate snRNP assembly and nuclear import. The western blot for SmB and SmD3 across cellular fractions demonstrated a largely nucleoplasmic localization of both proteins in all tested conditions that increased following Type II PRMT inhibition.

Reviewer #1 (Recommendations for the authors):1) Show deltaPSI in Figure 1e not log(PRMT/DMSO) to be consistent and to clearly indicate the differences seen.

We agree that ΔPSI is a valuable metric for transcriptome-wide studies. However, we argue that the log2 representation is informative and the standard approach in this context of analysis of specific introns. log2 transformations are generally expected to be normally distributed and therefore allow us to use a t-test. Therefore, for the RT-qPCR analyses we maintained the use of log2 transformations and for RNA seq we used ΔPSI.

2) I don't understand the rationale of the statement "a slower splicing rate should increase RI" (line 188). Isn't it possible that a slower rate increases fidelity and thus increases intron removal at steady state, whereas rapid splicing might "miss" some introns? Maybe I'm wrong, but I'm not sure it is fair to conclude that splicing rate leads to the conclusion that regulation by type I PRMTs is post-transcriptional.

The primary indication that these introns are regulated post-transcriptionally is that the splicing rate is not inversely changed with PRMT inhibitors despite the changes seen in the poly(A) RNA. We have edited that paragraph to read:

“…As we initially hypothesized that a slower splicing rate should increase RI, this result was surprisingly inconsistent with the PRMT-inhibited changes seen in poly(A) RNA (Figure 1b). These observations therefore suggested that changes in intron retention following PRMT inhibition were not entirely due to co-transcriptional splicing.”

3) In the Sm mutational experiments, are the R-A mutants incorporated into snRNPs? That is, is the impact due indirectly to a decreased number of snRNPs or a direct effect on the activity of the snRNPs?

This is a very interesting question. We performed co-immunoprecipitation experiments targeting the FLAG epitope located at the N-terminus of the Sm expression construct. Indeed, we enriched SMN and U1-70k, as seen by western blot. We also performed a northern blot for the major snRNAs (U1, U2, U4, U5, U6) and detected all of them. We also performed a SNRPB/SmB IP followed by northern blotting for the major snRNAs to detect both endogenous and exogenous Sm protein assembly into snRNPs. We similarly observed intact snRNP assembly. We performed this experiment with the PRMT inhibitors and saw unperturbed snRNP assembly. Indirect immunofluorescence microscopy confirmed that exogenous Sm proteins are imported into the nucleus. Therefore, we now conclude in the manuscript text that snRNP assembly is unaffected by PRMT inhibition. Importantly, these results are consistent with prior literature indicating that deletion of the *Drosophila* PRMT5 orthologue does not affect snRNP assembly (1) and that deletion of the Sm tails also does not perturb their transfer onto the SMN complex (2).

1. G. B. Gonsalvez, T. K. Rajendra, L. Tian, A. G. Matera, The Sm-Protein Methyltransferase, Dart5, Is Essential for Germ-Cell Specification and Maintenance. Current Biology 16, 1077-1089 (2006).

2. A. Chari et al., An Assembly Chaperone Collaborates with the SMN Complex to Generate Spliceosomal SnRNPs. Cell 135, 497-509 (2008).

Reviewer #2 (Recommendations for the authors):Figure 1: The inverse splicing of many RIs in response to the different inhibitors is evident in 1c, but these numbers add up to less than half of the introns affected by MS023 shown in f and g. If you measure these exons as a population (rather than comparing them one-for one) in the GSK591 treatment vs. the MS023 treatment, does the distribution move significantly in the opposite direction? The question is whether this subset is responding in the same inverse manner but is below significance threshold, or whether these may represent some indirectly affected second group.

To address this interesting critique, we isolated the common introns in all conditions and tested their median ΔPSI. This did not significantly change the median ΔPSI from the total population, so we consider them appropriately representative for the subsequent analysis. We added the following sentence to the manuscript:

“When analyzing the distribution of ΔPSI for this subset of common introns (as in Figure 1b), the median ΔPSI did not significantly differ from the parent population (not shown). This observation indicates that this subset of RI is suitably representative of the larger group of PRMT-regulated RI.”

Figure 2: This figure is difficult to follow, although along with Figure 3 makes the important observation that these effects are most likely post-transcriptional, given that the expected result for co-transcriptional effects would be the opposite of what is seen with MS023 treatment. Specifically, 2 c and d and 3 e are the critical pieces of data. Rearranging the order of the figure panels and potentially moving the more trivial results into supplemental might help clarify the presentation.The measurement of splicing rates and the here is straightforward and directly determined by the experimental data. I'm less convinced that the PSI simulation in panel b is not a circular process, and more direct experimental data is required to make the argument that alternative splicing ratios are so universally and profoundly affected by elongation rates.

To clarify this figure and make it easier for the reader to reach the conclusion that RI regulation is likely post-transcriptional, panel 2b was moved to Figure 2 —figure supplement 1 and panels 2f and 2g were removed. Panel 3a was moved to Figure 2 —figure supplement 2 and panel 3b was added to Figure 2. Furthermore, we analyzed the splicing rate, time to transcribe, and probability of splicing prior to cleavage for the RI present in GSK591 and MS023 in the DMSO-treated cells as well (now included in Figure 2). This analysis revealed the same characteristics as seen with GSK591 or MS023 treatment: a slower splicing rate, faster time to transcribe, and higher probability of cleavage prior to splicing. Together, these results—in addition to the lack of changes in co-transcriptional kinetics following PRMT inhibition—support that a post-transcriptional mechanism is at play.

The fact that the data in panels g and h are contradictory to the elongation rate model of splicing (since splicing rates should be determined by the overall time to transcribe, which is significantly faster here, even though measured elongation and splicing rates are lower, in both treatments) supports the conclusion that the differential effects of PRMTi occurs post-transcriptionally. The current presentation of the data in the text, especially the concluding sentence (line 211 pg. 11) "Taken together, these results support the conclusion that RI in GSK591- and MS023-treated cells share unique characteristics that increase their probability of post-transcriptional processing" should more explicitly state that the unique characteristics referred to are that they don't behave according to the model-as written, the reader spends a lot of time trying to figure out how these seemingly contradictory data support the conclusion.

We addressed this critique by clarifying and expanding the writing in the main text.

Figure 3: This is a difficult experiment, but really pins down the post-transcriptional effect on intron abundance.

Thank you.

Figure 4: It would be potentially very informative to see the soluble (nucleoplasmic) and cytoplasmic fractions of these blots-is there a pool of methylated SNRPB in steady-state that moves to the chromatin fraction upon demethylation (in other words, is the methylation state likely controlling the localization of the protein). The same fractionation on the arginine substitution mutants would further address the question.

We thank the reviewer for this critique as it prompted us to further explore the localization issue. As shown in new Figure 6c,d,e, besides increased Sm proteins in GSK591-treated chromatin and nucleoplasmic fractions, there were no gross differences in Sm and CHTOP subcellular localization upon PRMT inhibition. Unfortunately, due to time constraints of the lead author’s return to medical school, and due to the difficulty in expanding the arginine substitution mutants to a large enough culture to perform fractionation, we did not perform the last suggested experiment.

Figure 7: The only part of the model that is not directly supported by the data here is the inhibition of RNA export and the promotion of post-transcriptional splicing via PRMT1 inhibition. If the observation were of spliced mRNA remaining nuclear in the presence of PRMT1 inhibition, that pathway could be inferred. However, in the absence of export, even fully-spliced mRNA might be turned over quickly. Alternatively, PRMT1 inhibition could result in a failure to protect DI-containing transcripts, which are then turned over in the nucleus. Dissecting this part of the pathway will likely require further experiments involving nuclear exosome inactivation.

We thank the reviewer for these intriguing suggestions. We further discussed these issues in the text.

Reviewer #3 (Recommendations for the authors):I have several concerns that I feel need to be addressed before settling on these interpretations, including details of particular protein activities invoked, e.g. by CHTOP, where the experimental procedures do not necessarily match the question posed and/or the conclusions drawn.1. The manuscript is really poorly written and hard to understand. The language is unnecessarily complicated and involves a lot of double negatives that seem unnecessary. For example, lines 15-18 in the abstract are literally impossible to understand. On line 22 "Conversely" is not understandable, because the results do recapitulate the splicing changes seen (as do the results in the previous sentence); "similarly" or "in addition" might be appropriate. This type of confusing language choice is pervasive.

We improved the writing throughout the manuscript. Due to the complicated nature of the independent and opposite contributions of Type I and II PRMTs to splicing regulation, in several places in the manuscript we expanded explanations into multiple sentences. We changed line 22 to “Similarly”.

2. The use of the term PRMTi obscures clarity about the specific experiments, which are done mostly with chemical inhibitors. In fact, the sentence "We also examine which factors are responsible for PRMTi-mediated splicing consequences" (lines 75-76) suggests PRMTi is a normal physiological pathway, which it is not. I strongly suggest the authors remove PRMTi as a term used in the paper. A mostly supplemental section uses a CRISPR interference strategy taken for validation, which was less robust that chemical inhibitor treatments. The authors must investigate the effects of chemical inhibition and CRISPR interference methods on dimethylarginine levels in order to correctly draw their conclusions! Other labs have reported only modest effects on sDMA and aDMA after even longer incubations.Related to the above issue of language, the statement on lines 301-2 "…. this suggested that SNRPB may also be involved in the regulation of GSK591-mediated RI." This is so confusing for the reader. Its sounds like SNRPB is an alternative splicing factor involved in regulating something that is not physiological. What does this sentence mean?

We removed the term PRMTi from the paper and replaced it with detailed inhibitor references. We also confirmed cellular methylarginine levels by western blot analysis (Figure 1 —figure supplement 1) and reference our recently published paper on mass spec analysis of the methylarginine proteome in the same A549 cells (Maron et al. 2021, iScience). While the changes in total methylarginine levels presented in our new figure after two days of drug treatment are more modest than in the 7-day treatment we performed in the recent publication (Maron et al. 2021, iScience), they are present and are specifically found on Sm proteins as indicated.

We revised the indicated sentence to “Together with the fundamental role of snRNPs in assembling the spliceosome, this suggested that SNRPB methylation may regulate intron retention.”

3. GSK591 inhibits PRMTs 5 and 9. It is stated on line 90 that "PRMT5 is the primary Type II methyltransferase (Yang et al., 2015)", and Type II is dropped from discussion for the remainder of the manuscript. For example, the figure title for Figure 1 is "Type I PRMTs and PRMT5 inversely regulate intron retention". Note that Yang et al. show that PRMT9 methylates SAP145 (SF3B2) and that knockdown affects splicing; SAP145 is a protein component of the U2 snRNP that is required for A complex formation. Therefore, ignoring PRMT9 is not justifiable. The authors should refer to Type II PRMTs throughout the text and in the title of the paper. As it stands, the title is not justified by the data.

We have retitled the manuscript and included Type II (as opposed to PRMT5) in the text. We are, however, unaware of any published evidence that GSK591 inhibits PRMT9. All publications and reviews describing GSK591 only document its inhibition of PRMT5.

4. Post-transcriptional splicing is investigated by inhibiting transcription in Figure 3. It is becoming increasingly clear that wholesale chemical inhibition of transcription or splicing can lead to a generalized stress response that may not reflect the "normal" unstressed situation. Would it not be possible to conduct their experiment in nucleoplasm versus chromatin, as was done by others and analyzed in Figure 6? This is especially frustrating, given the authors do do the fractionation in the context of figure 4.

In the revised manuscript, we performed cellular fractionation in the presence of the PRMT inhibitors with and without actinomycin D treatment. This experiment demonstrated that the intron decay for our candidate introns is localized to the nucleoplasmic and chromatin fractions—consistent with nuclear detention (new Figure 6e). We also tried alternative approaches using 5-ethynyl uridine labeling of nascent RNA; however, these experiments had large technical hurdles. We hope to optimize and implement such approaches for future study.

5. In figure 4, the authors identify a post-translational mRNA interactome by isolating polyA+ RNA from chromatin. Overall, this is a really great undertaking with potentially high impact. However, few of the dots are connected for us in the context of this experiment. The rationale is that previous studies suggest retention of transcripts on chromatin. The interpretation of this figure is that PRMT inhibitors shift proteins aberrantly in and out of the chromatin-proximal polyA+ pool. To make this interpretation, it would be necessary to analyze all of the polyA+ interacting proteins in nucleoplasm and compare them with the pool of interactors in the chromatin fraction. Otherwise, other less direct interpretations could come into play. It is unclear how specific the polyA+ RNA isolated in this experiment is; how can we distinguish between background binding and polyA+ RNA specifically retained on chromatin due to continued processing? It would be great if the authors undertook some orthogonal experiments, e.g. through imaging, to characterize the localization patterns of snRNPs and CHTOP relative to one another and to chromatin/nascent transcripts in the presence and absence of the inhibitors.

We recognize the reviewer’s interest in our important experiment and that understanding how arginine methylation regulates the distribution of proteins between the nucleoplasm and chromatin fractions is important. We addressed this critique by probing the cytoplasmic, nucleoplasmic, and chromatin fractions by western blotting (shown in new Figure 6c). We and others hope to address this with higher resolution in our future work.